# Autophagy drives the conversion of developmental neural stem cells to the adult quiescent state

Isabel Calatayud-Baselga[1], Lucía Casares-Crespo[1], Carmina Franch-Ibáñez [1], José Guijarro-Nuez[1], Pascual Sanz [1] & Helena Mira [1] ✉

Neurogenesis in the adult mammalian brain relies on the lifelong persistence of quiescent neural stem cell (NSC) reservoirs. Little is known about the mechanisms that lead to the initial establishment of quiescence, the main hallmark of adult stem cells, during development. Here we show that protein aggregates and autophagy machinery components accumulate in developmental radial glia-like NSCs as they enter quiescence and that pharmacological or genetic blockade of autophagy disrupts quiescence acquisition and maintenance. Conversely, increasing autophagy through AMPK/ULK1 activation instructs the acquisition of the quiescent state without affecting BMP signaling, a gatekeeper of NSC quiescence during adulthood. Selective ablation of *Atg7*, a critical gene for autophagosome formation, in radial glia-like NSCs at early and late postnatal stages compromises the initial acquisition and maintenance of quiescence during the formation of the hippocampal dentate gyrus NSC niche. Therefore, we demonstrate that autophagy is cell-intrinsically required to establish NSC quiescence during hippocampal development. Our results uncover an important role of autophagy in the transition of developmental NSCs into their dormant adult form, paving the way for studies directed at further understanding the mechanisms of stem cell niche formation and maintenance in the mammalian brain.

Neurogenesis continues throughout life in the adult mammalian brain thanks to neural stem cells (NSCs) persisting in defined niches beyond development. These stem cells ultimately derive from proliferating developmental precursors that are set aside as reservoirs, either embryonically in the ventricular-subventricular zone (V-SVZ) niche[1,2] or postnatally in the hippocampal dentate gyrus subgranular zone (SGZ) niche[3,4]. Acquisition of quiescence (a reversible exit from the cell cycle) while maintaining stemness is considered a critical turning point that defines the conversion of developmental stem cells to their adult counterparts[5,6]. In adulthood, quiescence is maintained by dominant extrinsic signals produced by various mature niche cell types[7,8]. A main regulator of adult stem cell quiescence is canonical bone morphogenetic protein (BMP) signaling through the BMPR-IA receptor expressed

in radial glia-like adult NSCs[9]. Since NSC quiescence is initially established before a fully functional adult niche structure is assembled, it is currently unknown how NSCs first come to rest[6].

Using the hippocampal dentate gyrus as a model system, we hypothesized that cell-intrinsic mechanisms, including those related to proteostasis, could contribute to quiescence entry during development. Proteostasis is key for the fitness of long-lasting cells like stem cells[10]. It controls protein turnover through intertwined pathways regulating protein synthesis, folding and degradation through the ubiquitin–proteasome system (UPS) or the autophagy–lysosomal pathway (ALP). Previous studies focused on adult brain stem cells reported lower protein synthesis, protein aggregate accumulation and higher lysosomal activity in adult quiescent NSCs[11–14]. This suggests

[1]Instituto de Biomedicina de Valencia, Consejo Superior de Investigaciones Científicas (IBV-CSIC), València, Spain. ✉e-mail: hmira@ibv.csic.es

that ALP could be playing an additional, unexplored, cell-autonomous function in driving the initial transition of developmental NSCs into their adult form. To tackle this question, in this work, we focus specifically on the role of autophagy in the acquisition of quiescence by radial glia-like NSCs located in the hippocampal SGZ niche, a poorly understood developmental transition. Here, we show that protein aggregates and autophagy machinery components accumulate in radial glia-like NSCs as they enter quiescence during postnatal development and that blockade of autophagy disrupts the initial acquisition and maintenance of the NSC quiescent state.

## Results

Embryonic dentate progenitors migrate from the neuroepithelium to the primitive dentate gyrus, forming the definitive adult radial glia-like NSC (RGL) reservoir during early postnatal development[3,4,15]. After a proliferation peak at postnatal day 3 (P3), RGLs occupy their final position in the SGZ at around P14 and continue to enter quiescence until P21[3,4] (Supplementary Fig. 1 a–d). We first set out to measure overall proteostatic events in P3 and P21 hippocampal NSCs at the single-cell level. NSCs were acutely isolated by FACS on the basis of the combined expression of the cell surface markers GLAST and Prominin-1[16,17] (Fig. 1a, Supplementary Fig. 2a). We compared GLAST+PROM-1+ cells at both ages. P21 NSCs substantially accumulated protein aggregates (visualized by the trapping of the fluorescent molecular rotor ProteoStat; Fig. 1a, b), a finding corroborated in hippocampal GLAST+ cells isolated by MACS (Supplementary Fig. 2b–d). At P21, proliferation of GLAST+ cells (percentage of Ki-67+ cells) markedly decreased (Fig. 1c) and global protein synthesis (OP-Puro signal) was reduced (Fig. 1d), but protein aggregates (ProteoStat signal, Fig. 1e) and lysosomes (LAMP2 signal, Fig. 1f) accumulated, pointing to a shift in hippocampal RGL proteostasis during the P3 (proliferation) to P21 (quiescence) developmental transition. Likewise, quiescent P21 and P14 RGLs identified in vivo as SOX2/GFAP co-expressing cells with radial morphology that were negative for the cell cycle marker Ki-67, showed increased LAMP2 levels compared to actively dividing (Ki-67+) RGLs (Fig. 1g, h). LAMP2 content reflected the proliferative state of the cells since LAMP2-low RGLs were actively proliferating while LAMP2-high cells were quiescent.

To explore the functional significance of these observations, we moved to an in vitro system that allows the controlled induction of quiescence in hippocampal neural stem and progenitor cell (NSPC) cultures, employing the quiescence-promoting signal BMP4[9,18] (Fig. 2a). In line with the FACS/MACS data, protein aggregates accumulated in quiescent (Q) NSPCs compared to active (A) NSPCs (Fig. 2b, c). When quiescence was reverted by BMP4 removal (Qrev), protein aggregate levels partially returned to those found in the A state (Fig. 2b). We then analyzed the activity of the two main branches of protein catabolism (UPS and ALP) to investigate if aggregate accumulation was due to a protein degradation failure. We found no differences in proteasome activity between active and quiescent NSPCs (Δ, Fig. 2d). However, Q-NSPCs accumulated LC3-II and SQSTM/p62 (autophagy markers; Fig. 2e, f and Fig. 2i, j) along with Tax1bp1 (aggrephagy receptor; Fig. 2h) and LAMP2 (lysosomal marker; Fig. 2l). Thus, we next evaluated autophagy flux, tracking the LC3-II increase upon disruption of autophagosome–lysosome fusion and lysosome acidification with Bafilomycin A1 (BafA1)[19]. The LC3-II/β-actin increase in BafA1-treated cells vs. BafA1-untreated cells was similar in both cellular states despite a higher initial LC3-II level in Q-NSPCs (Fig. 2f; see also Supplementary Fig. 3a, b for LC3-II levels normalized to total protein). As shown in Fig. 2g, the autophagy flux expressed as the ratio of LC3-II(+BafA1)/LC3-II(-BafA1) was equivalent for A-NSPCs and Q-NSPCs. Enhanced LC3-II levels in Q-NSPCs accounted for an increase in autophagic vesicles, since quantification of the signal in the presence of the selective unc-51-like autophagy activating kinase 1 (ULK1) inhibitor (SBI-0206965) showed that lipidated LC3-II accumulation

was dependent on the initiation of the autophagy pathway (Supplementary Fig. 3c, d). Similar results were obtained when SQSTM/p62 dynamics were evaluated as a proxy for the autophagy flux (Fig. 2k; see also Supplementary Fig. 4 for p62 levels normalized to total protein and for p62 levels quantified per cell by immunofluorescence). To further assess the transition from autophagosomes to autolysosomes, we electroporated the NSPCs with a pRFP-GFP-LC3 tandem sensor[20], which expresses the red fluorescent protein (RFP) fused to LC3 and a pH-sensitive green fluorescent protein (GFP). This sensor allows autophagosomes to be identified as yellow (green and red) cytoplasmic puncta, whereas autolysosomes are visualized as red puncta, having lost the GFP signal (Fig. 2m). In line with our previous results, Q-NSPCs had more GFP+RFP+ autophagosomes than A-NSPCs after BafA1 treatment, but a similar autophagic flux (Fig. 2m, Supplementary Fig. 5). These findings suggest that the increase in ALP markers in quiescent cells is likely due to an increase in autophagy machinery content, rather than due to a deficient autophagic flux or an increase in non-autophagic membranes related to pathways such as LC3-associated endocytosis. The rise in autophagy machinery possibly occurs in turn as a compensatory response to the accumulation of protein aggregates.

To confirm this, we next explored the overall expression profile of ALP-related genes in Q-NSPCs and A-NSPCs by bulk RNAseq. In Q-NSPCs, we found a clear enrichment in genes ascribed to the Macroautophagy and Lysosome Gene Ontology (GO) biological functions (Fig. 3a, Supplementary Data 1). The gene set partly overlapped with ALP-related genes upregulated in *Hopx*-expressing dentate gyrus precursors when they transition from the proliferative state (P4) to the adult quiescent state[4] (Fig. 3b, Supplementary Data 1). It was also shared in part by Q-NSPCs acutely isolated from the adult V-SVZ niche[11] (Supplementary Fig. 6a). A selection of genes was further validated by RT-qPCR in independent samples (Fig. 3c), including genes involved in the initial induction steps of the autophagy pathway (ULK1-complex genes and the mTOR repressor *Tsc1* gene, Fig. 3c, and AMPK genes, Supplementary Fig. 6b) and key downstream components such *Atg7*, whose product is critically required for generating lipidated LC3-II that in turn participates in autophagosome formation (Fig. 3c). AMPK senses the energy status (AMP:ATP ratio) of the cell and is a major upstream positive regulator of autophagy[21]. AMPK was phosphorylated at Thr172 and was thus activated in Q-NSPCs (Fig. 3d). Phosphorylation of its target ULK1 at Ser555 was increased in Q-NSPCs (Fig. 3e). Moreover, raptor phosphorylation at Ser792 was also enhanced (Supplementary Fig. 6c), suggesting an inhibition of the mTOR pathway (a negative regulator of autophagy), further demonstrating that Q-NSPCs are engaged in autophagy induction. Together, the data suggest that Q-NSPCs accumulate protein aggregates of unknown origin and that this is not due to arrested protein degradation. Rather, Q-NSPCs activate AMPK/ULK1 and engage in a transcriptional programme that enhances the autophagy machinery to deal with the protein aggregates.

Next, we designed a series of in vitro experiments to evaluate if autophagy had a positive impact on the entry and maintenance of NSPC quiescence. We employed both pharmacological activators and inhibitors of autophagy and siRNA-mediated silencing of the SQSTM/p62 autophagy receptor. Proliferative cells were identified based on Ki-67 immunostaining (Fig. 4). The AMPK activator phenformin (Phen; Supplementary Fig. 6d) favored cell cycle exit in NSPC cultures treated with the quiescence inducer BMP4 (Fig. 4a; see Supplementary Fig. 7a for similar proliferation results based on BrdU incorporation). On the contrary, decreasing SQSTM/p62 levels by siRNA impaired quiescence acquisition (Fig. 4b). Most importantly, AMPK activation with Phen blocked proliferation of A-NSPCs grown in the presence of the mitogen FGF2 and was sufficient to induce cell cycle exit even in the absence of the quiescence-promoting signal (Fig. 4c). Since AMPK has multiple functions beyond regulating autophagy, we employed the ULK1 kinase

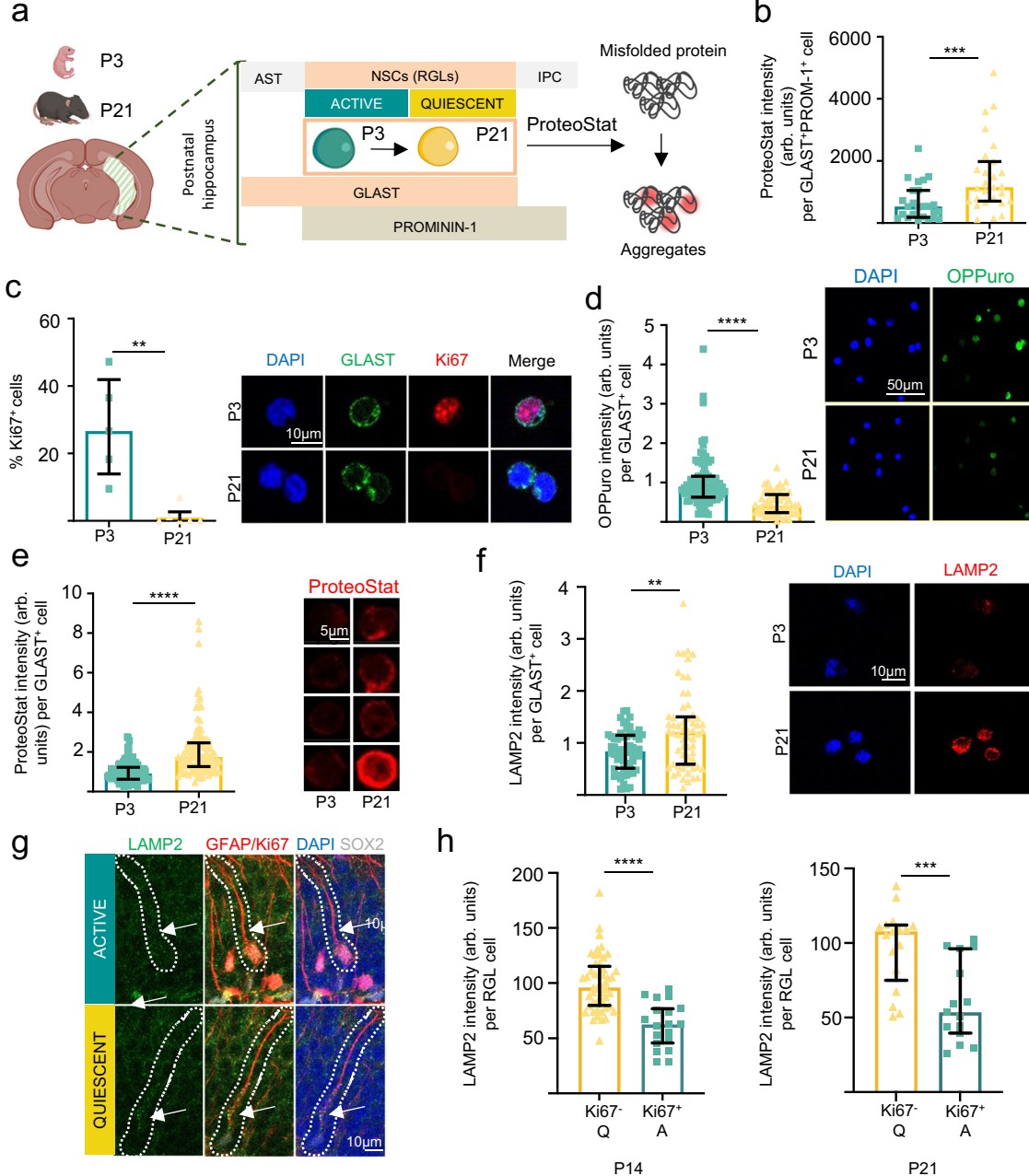

**Fig. 1 | Protein aggregates and lysosomes accumulate in quiescent P21 hippocampal NSCs compared to proliferating P3 NSCs. a** Experimental design. ProteoStat labeling of protein aggregates was performed on GLAST⁺Prominin-1⁺ NSCs isolated by FACS from the hippocampus of mice on postnatal day 3 (P3) and P21. **b** Mean fluorescence of ProteoStat per GLAST⁺Prominin-1⁺ per cell, measured by confocal microscopy. Each point represents an analyzed cell (*n* = 25 and 27 cells, P3 and P21, respectively). **c** (Left) Percentage of proliferative cells (Ki67⁺) cells isolated from P3 and P21 mice. (Right) Representative immunocytochemistry images of GLAST/Ki67 in GLAST⁺ cells isolated from postnatal 3 and 21-day-old mice (*n* = 5 independent isolations of 8 P3 mice and *n* = 6 independent isolations of 6 P21 mice were used). Scale bar: 10 μm. **d** (Left) Quantification of OPPuro fluorescence in GLAST⁺ cells isolated from P3 and P21 old mice. (Right) Representative immunofluorescence confocal images of GLAST⁺ cells isolated from P3 and P21 mice treated with OPPuro, which is incorporated in newly synthesized proteins (*n* = 91 and 54 cells, P3 and P21, respectively). Scale bar: 50 μm. **e** (Left) Quantification of protein aggregates measured with ProteoStat in GLAST⁺ cells. (Right) Representative immunofluorescence confocal images of GLAST⁺ cells isolated from P3 and P21 mice stained with ProteoStat (red) (*n* = 119 and 103 cells, P3 and P21, respectively).

Scale bar: 5 μm. **f** (Left) Quantification of LAMP2 in GLAST⁺ cells. (Right) Representative immunofluorescence confocal images of GLAST⁺ cells isolated from P3 and P21 mice stained with LAMP2 (*n* = 56 and 60 cells, P3 and P21, respectively). Scale bar: 10 μm. **g** (Left) Representative immunohistochemistry confocal images of LAMP2 (green) in active and quiescent RGLs labeled for GFAP (red), SOX2 (white) and Ki67 (red). Scale bar: 10 μm. **h** Quantification of LAMP2 intensity in active (Ki67⁺) or quiescent (Ki67⁻) RGLs (GFAP⁺/SOX2⁺) in the dentate gyrus of postnatal 14 (P14, left) and 21 (P21, right) days old wild type mice (3 mice per postnatal age were used). Each point represents an analyzed RGL cell (*n* = 49 and 19 cells, Q and A P14, respectively and *n* = 17 and 15 cells, Q and A P21, respectively). Data in panels (**b**–**h**) are presented as median ± interquartile range from at least three independent isolations. Statistics in panels (**b**–**h**): Mann–Whitney test, two-tailed. 3 biological replicates of 8 P3 and 6 P21 mice were performed in panels (**b, d, e** and **f**). *$p < 0.05$; **$p < 0.01$; ***$p < 0.001$; ****$p < 0.0001$. Representation shown in (**a**). was created with BioRender.com. Source data are provided as a Source Data file. AST Astrocyte, NSCs Neural Stem Cells, RGLs Radial Glia-Like Cells, IPC Intermediate Progenitor Cell, P3 Postnatal day 3, P21 Postnatal day 21, Q quiescent, A active.

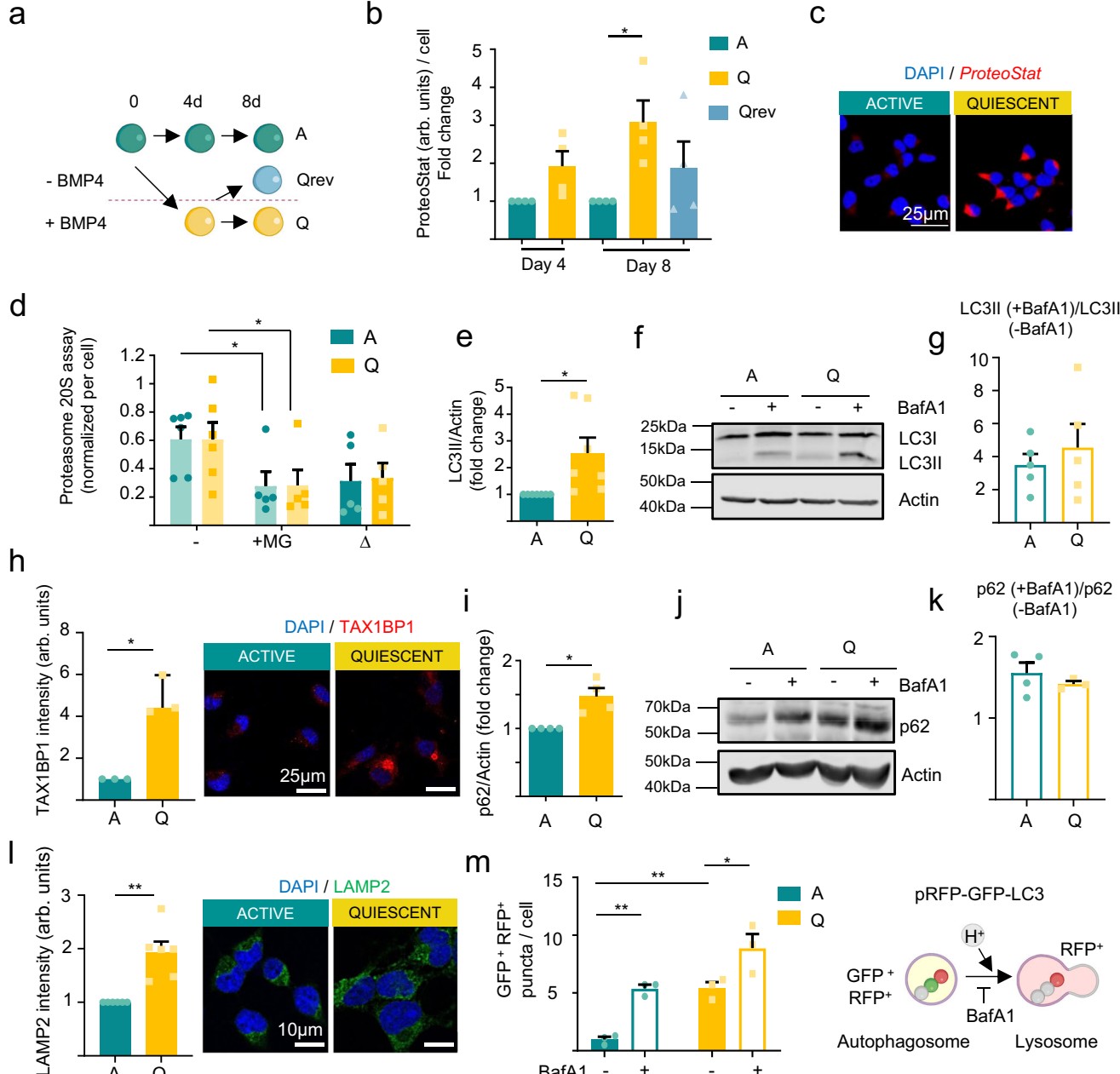

**Fig. 2 | Protein aggregates accumulate in quiescent NSPC cultures despite the increase in autophagy machinery content. a** Experimental design. Hippocampal NSPC cultures are grown in the presence of fibroblast growth factor 2 (FGF2) and quiescence is induced by the addition of bone morphogenetic protein 4 (BMP4). Removing BMP4 reverses quiescence. **b** Quantification of protein aggregates measured by ProteoStat labeling at 4 and 8 days of quiescence induction and 4 days of quiescence reversion. At least 15 cells were analyzed per experiment and condition (*n* = 4 independent experiments). Statistics: one-sample *t*-test, two-tailed. **c** Representative confocal immunofluorescence images of Q-NSPC and A-NSPC cultures stained with ProteoStat (red). Scale bar: 25 μm. **d** Quantification of proteasome 20S chymotrypsin activity (Δ) in A- and Q-NSPCs upon its inhibition using 10 μM MG132 (*n* = 5 independent experiments for MG132 treated cells and *n* = 6 independent experiments for control untreated cells). Statistics: two-way ANOVA. **e** Quantification of LC3-II levels in A- and Q-NSPCs (*n* = 7 independent experiments). Statistics: one-sample *t*-test, two-tailed. **f** Immunoblots of LC3 in A- and Q-NSPCs after 100 nM BafA1 treatment (6 h). β-actin was used as a loading control. **g** The autophagy flux expressed as the ratio of LC3-II(+BafA1)/LC3-II(-BafA1) was equivalent for A-NSPCs and Q-NSPCs. **h** (Left) Quantification of TAX1BP1 protein level measured as fluorescence intensity. Each point represents individual cultures (*n* = 3 independent experiments). Statistics: one-sample *t*-test, two-tailed. (Right)

Representative immunofluorescence confocal images of TAX1BP1 in A- and Q-NSPCs. Scale bar: 25 μm. **i** Quantification of p62 levels in A- and Q-NSPCs (*n* = 4 independent experiments). Statistics: one-sample *t*-test, two-tailed. **j** Immunoblots of p62 in A- and Q-NSPCs after 100 nM BafA1 treatment (6 h). β-actin was used as a loading control. **k** The autophagy flux expressed as the ratio of p62(+BafA1)/p62(-BafA1) was equivalent for A-NSPCs and Q-NSPCs. **l** (Left) Quantification of LAMP2 protein level measured as fluorescence intensity. Each point represents individual cultures (*n* = 6 independent experiments). Statistics: one-sample *t*-test, two-tailed. (Right) Representative immunofluorescence confocal images of LAMP2 in A- and Q-NSPCs. Scale bar: 10 μm. **m** (Right) In electroporated cells expressing the pRFP-GFP-LC3 sensor, autophagosomes display GFP and RFP fluorescence (yellow puncta), whereas autolysosomes display RFP fluorescence only, because GFP is denatured at the acidic pH of the lysosome. (Left) Quantification of the number of autophagosome puncta (GFP⁺RFP⁺) in A- and Q-NSPCs after 100 nM BafA1 treatment (6 h) (*n* = 3 independent experiments). Statistics: two-way ANOVA. Data in panels (**b**, **d**, **e**, **g**, **h**, **i**, **k**, **l**, and **m**) are presented as mean ± SEM. \**p* < 0.05; \*\**p* < 0.01; \*\*\**p* < 0.001. Representations shown in (**a** and **n**) were created with BioRender.com. Source data and all blots are provided as a Source Data file. A, active NSPCs. Q, quiescent NSPCs. Qrev quiescence-reverted NSPCs.

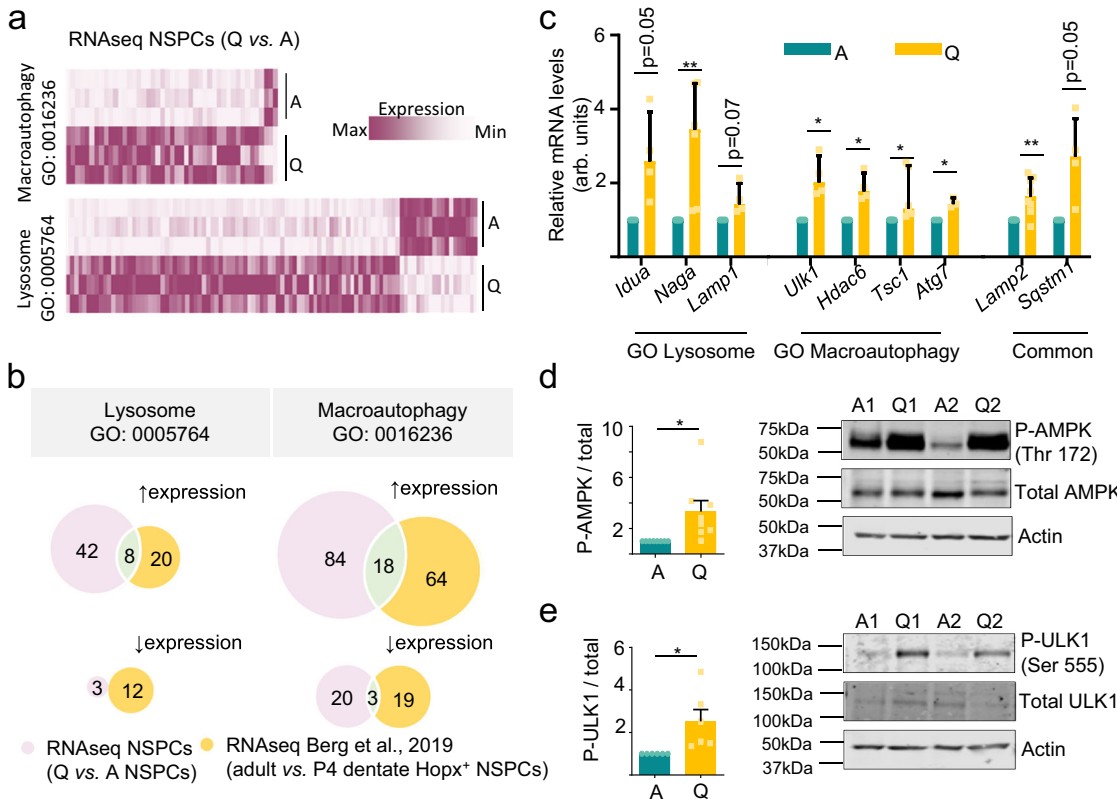

**Fig. 3 | Autophagy is induced in quiescent NSPCs. a** Heatmap showing the enrichment in Macroautophagy-GO:0016236 (top) and Lysosome-GO:0005754 (bottom) gene expression in Q-NSPCs versus A-NPSCs. Maximum expression: purple; minimum expression: white. Data were obtained by bulk RNAseq. **b** Venn diagrams identify common genes associated with Lysosome-GO (Left) and Macroautophagy-GO (Right) in the RNAseq data from this study and RNAseq data from Berg et al., 2019, corresponding to NSCs isolated from 4-day-old (P4) and adult *Hopx-CreER^T2^::EYFP* mice. NSCs from adult mice are mostly quiescent; P4 NSCs are mostly active. **c** Relative expression of autophagy–lysosome genes in A- and Q-NSPCs by RT-qPCR analysis (*n* = 3, 4, 5 or 8 independent experiments). **d** Left:

quantification of phosphorylated P-AMPK (Thr172) levels in A- and Q-NSPCs. Right: representative immunoblots of total AMPK and P-AMPK (Thr172) in A- and Q-NSPCs. β-actin was used as a loading control. **e** (Left) Quantification of phos-phorylated (P) ULK1 (Ser555) levels in A- and Q-NSPCs (*n* = 6 independent experiments). (Right) Immunoblots of total ULK1 and P-ULK1 (Ser555) in A- and Q-NSPCs. β-actin was used as a loading control. Data in panels (**c**–**e**) are presented as mean ± SEM except for *Tsc1* in (**c**), represented as median ± interquartile range. Statistics in panels (**c**–**e**): one-sample *t*-test, two-tailed, except for *Tsc1* in (**c**) (Wil-coxon Signed Rank Test). *$p < 0.05$; **$p < 0.01$. Source data and all blots are pro-vided as a Source Data file. A, active NSPCs. Q, quiescent NSPCs.

inhibitor (SBI-0206965) to clarify if the Phen effect was dependent on the initiation of the autophagy pathway. Our results demonstrated that instruction of the quiescent state upon increasing AMPK activity was ULK1-dependent (Fig. 4c). ULK1 inhibition by itself did not raise pro-liferation of the A-NSPCs (Fig. 4c). Interestingly, promotion of the quiescent state through autophagy induction did not require canoni-cal BMP signaling. As shown in Supplementary Fig. 7b–e, the effect of the highly selective BMPR inhibitor DMH1 (Dorsomorphin Homolog 1) on the acquisition of NSPC quiescence induced by Phen was not sig-nificant, although it blocked cell cycle exit following BMP4 stimulation. Phosphorylation of the canonical BMP effector proteins SMAD1/5/9 in NSPCs and their nuclear translocation was unaffected by autophagy induction and inhibition (Phen and SBI treatments, respectively; Sup-plementary Fig. 8). The levels of type I and type II BMP receptors BMPR1A and BMPR2 were not significantly affected (Supplementary Fig. 8a, b). Thus, our results demonstrated that quiescence instruction by AMPK activation with Phen was not mediated by canonical BMP signaling. In addition, Phen prevented reactivation of the Q-NSPCs when BMP4 was removed (Fig. 4d), whereas the autophagy inhibitor BafA1 favored cell cycle entry (Fig. 4e).

Finally, we explored the in vivo requirement of autophagy in the establishment of the quiescent hippocampal SGZ neural stem cell reservoir during postnatal development, using a genetic approach. We designed a cell type- and time-specific strategy to accurately examine the role of autophagy in this process. We crossed mice expressing in

radial glia-like NSCs a tamoxifen (TX)-inducible form of Cre recombinase under nestin transcriptional control[22] (*NesCreER^T2^*), with conditional-knockout mice for the autophagy gene *Atg7*[23] (*Atg7^Fl/Fl^*; Fig. 5a and Sup-plementary Fig. 9a–f). To assess the early phase of postnatal SGZ niche development, tamoxifen or oil (control) was administered to P3-P4 animals to ablate *Atg7* in SGZ RGLs. Proliferating cells were labeled by BrdU injection at P5. Entry into quiescence of the actively dividing BrdU⁺ RGLs was evaluated at P14 (a critical timepoint when the morphological pattern and organization of the SGZ niche and DG subfields emerge[3,4]), using the cell cycle marker Ki-67 (Fig. 5a, upper panel). RGLs were identified throughout the study as SOX2/GFAP co-expressing cells with radial morphology (Fig. 5b). Active RGLs transitioning into quiescence were identified as BrdU⁺Ki-67⁻. We found a robust reduction in the number (2.8-fold) and percentage (1.7-fold) of proliferating P5 BrdU⁺ RGLs entering quiescence at P14 in tamoxifen-treated animals (BrdU⁺Ki-67⁻ RGLs, Fig. 5c, d). We also found a significant increase in the total number of P14 RGLs (Fig. 5e), suggesting that proliferating RGLs that fail to acquire dormancy expand through symmetric divisions in the time-frame analyzed. As a control, recombination of the targeted *Atg7* allele in the DG after tamoxifen administration was validated by PCR (Supple-mentary Fig. 9b). Loss of *Atg7* expression in radial glia-like NSCs at the timepoint of evaluation (P14) was corroborated at the mRNA level by RT-PCR of FACS isolated GLAST⁺PROM-1⁺ cells (Supplementary Fig. 9g, h).

We also analyzed the *Atg7* conditional-knockout mice at a later developmental timepoint, to evaluate the requirement of autophagy in

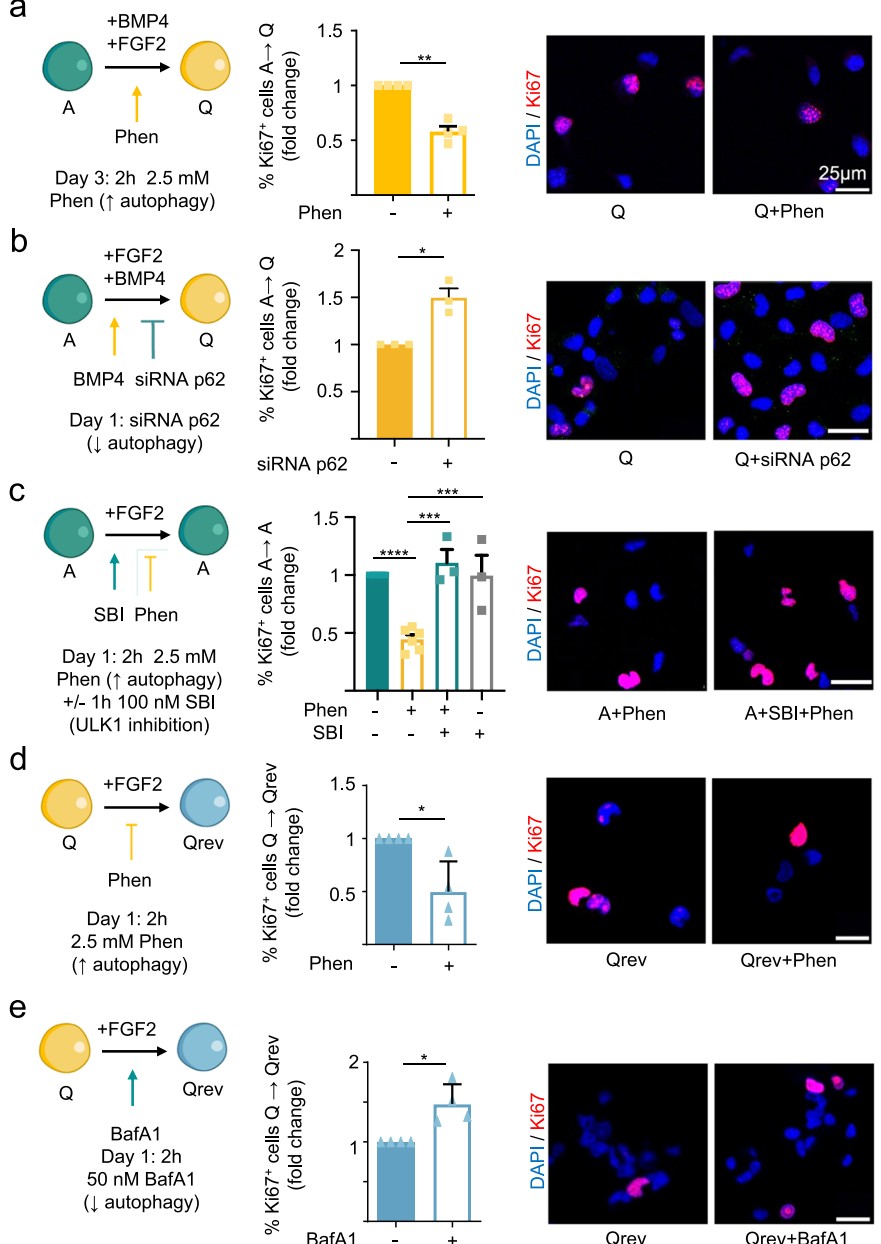

**Fig. 4 | Pharmacological modulation of autophagy regulates the switch between NSPC activation and quiescence. a** (Left) Functional assay scheme. Phenformin (Phen) activates the autophagy−lysosome pathway. A-NSPCs were treated for 2 h with Phen on Day 3 of quiescence induction. (Middle) Fold-change in the percentage of Ki-67+ cells 24 h after the 2 h Phen 2.5 mM treatment of A-NSPCs entering quiescence ($n = 4$ independent experiments). (Right) Representative confocal immunofluorescence images for Ki-67 (red) 24 h after Phen treatment. **b** (Left) Functional assay scheme. p62 siRNA prevents quiescence entry. (Middle) Fold-change in the percentage of Ki-67+ cells 72 h after p62 siRNA treatment of A-NSPCs entering quiescence ($n = 3$ independent experiments). (Right) Representative confocal immunofluorescence images for Ki-67 (red) 72 h after p62 siRNA treatment. **c** (Left) Functional assay scheme. A-NSPCs were treated with SBI, or DMSO, for 1 h and next Phen, or H$_2$O, was added for the following 2 h. Medium was then changed and cells were fixed 24 h later. (Middle) Fold-change in the percentage of Ki-67+ cells 24 h after the ±SBI (100 nM) and ±Phen treatment (2.5 mM) of A-NSPCs entering quiescence ($n = 3$ or 6 independent experiments). (Right)

Representative confocal immunofluorescence images for Ki-67 (red) 24 h after SBI and Phen treatment. **d** (Left) Functional assay scheme. (Middle) Fold-change in the percentage of Ki-67+ cells 48 h after the 2 h Phen (2.5 mM) treatment of Q-NSPCs that were being reverted to activation ($n = 4$ independent experiments). (Right) Representative confocal immunofluorescence images for Ki-67 (red) 48 h after Phen treatment. **e** Bafilomycin A1 (BafA1) inhibits the autophagy−lysosome pathway. (Left) Functional assay scheme. (Middle) Fold-change in the percentage of Ki-67+ cells 16 h after the 2 h BafA1 50 nM treatment of Q-NSPCs that were being reverted to activation ($n = 4$ independent experiments). (Right) Representative confocal immunofluorescence images for Ki-67 (red) 16 h after BafA1 treatment. Scale bars: 25 μm. Data in all panels are presented as mean ± SEM. Statistics: one-sample $t$-test, two-tailed, except (**c**): one-way ANOVA was performed. *$p < 0.05$; **$p < 0.01$; ****$p < 0.0001$. Representation shown in (**a**−**e**) were created with BioRender.com. Source data are provided as a Source Data file. A, active NSPCs. Q, quiescent NSPCs. Qrev, quiescence-reverted NSPCs. Phen, Phenformin. BafA1, Bafilomycin A1. SBI, SBI-0206965.

the maintenance of the quiescent stem cell reservoir upon the appearance of the SGZ niche organization. Tamoxifen or oil (control) was administered to P10−P13 animals to ablate *Atg7* in SGZ RGLs and proliferating cells were labeled by BrdU injection at P13. Entry into

quiescence of the actively dividing BrdU+ RGLs was evaluated at P21 using Ki-67 and MCM2 (Fig. 5a, lower panel). Similarly, quiescence maintenance was evaluated in BrdU− RGLs (Fig. 5a, b; Supplementary Fig. 10a, b). We found a substantial reduction in the number and

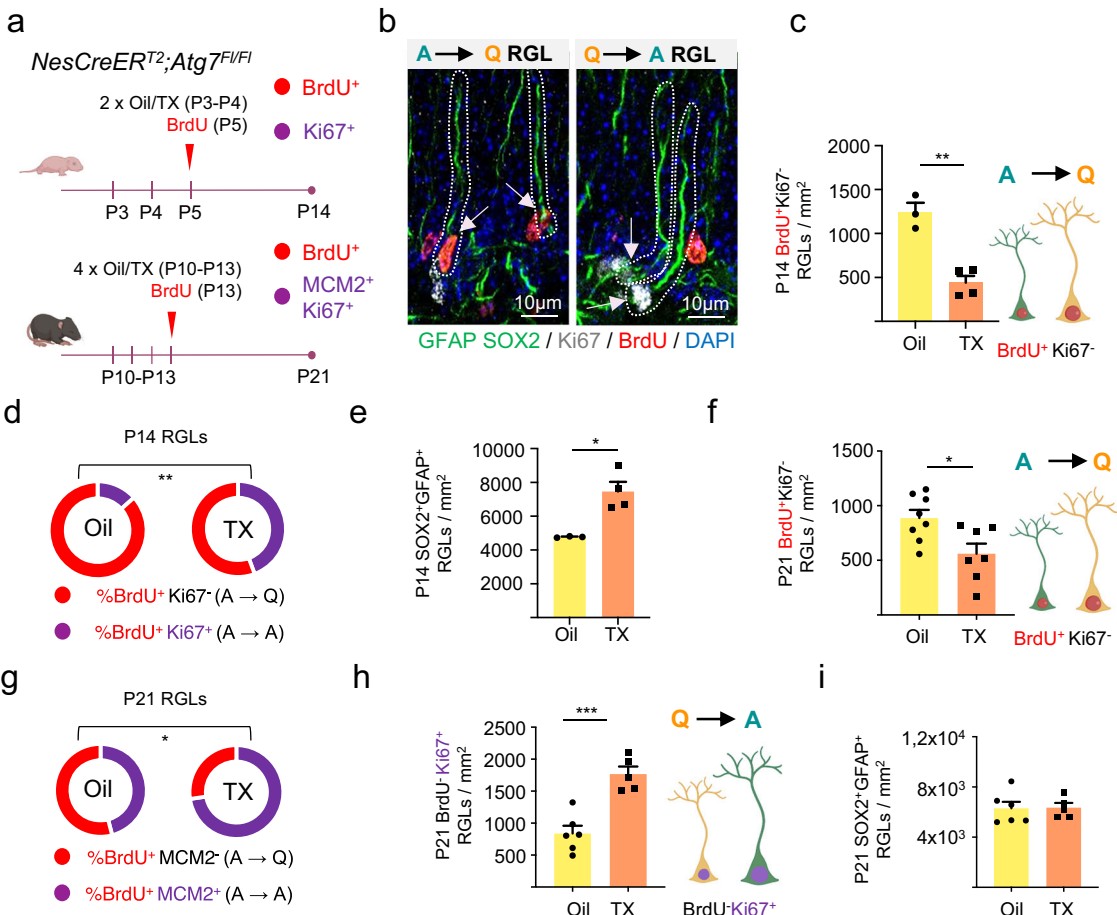

**Fig. 5 | The autophagy gene *Atg7* is cell-autonomously required for the establishment of hippocampal radial glia-like (RGL) neural stem cell quiescence during postnatal development. a** Schematic diagram of the experimental design. (Upper panel) *NesCreER^T2;Atg7^Fl/Fl* mice were injected with tamoxifen (TX) at P3–P4 to induce *Agt7* deletion in NSCs. Oil was used as control. Mice also received BrdU (Bromodeoxyuridine) injection to mark dividing cells at P5. Animals were sacrificed at P14 and proliferating cells were identified by Ki-67 staining. (Lower panel) *NesCreER^T2;Atg7^Fl/Fl* mice were injected with TX at P10–P13 to induce *Agt7* deletion in NSCs. Oil was used as control. The mice also received BrdU injection to mark dividing cells at P13. Animals were sacrificed at P21 and proliferating cells were identified by Ki-67 or MCM2 staining. **b** Double-labeling at the time of sacrifice enables the identification of active (A) RGLs (Radial Glia-Like Cells) that have entered quiescence (Q) (BrdU^+Ki-67^− A→Q RGLs, arrowheads in right panel) and quiescent RGLs that have entered the cell cycle (BrdU^−Ki-67^+ Q→A RGLs, arrowheads in left panel). RGLs were identified as SOX2^+ cells with their soma in the SGZ and a radial GFAP+ process crossing the granule cell layer. Representative confocal images of RGLs labeled for GFAP (green), SOX2 (green), Ki-67 (white) and BrdU (red) at P21 are shown. Scale bar: 10 μm. **c** Quantification of active RGLs entering quiescence (BrdU^+Ki-67^− RGLs/mm²) in the SGZ of *Atg7* conditional knockout (cKO; TX) and control (Oil) P14 mice (*n* = 3 and 4 mice, Oil and TX, respectively).

**d** Percentage of P5 active RGLs (BrdU^+) entering quiescence (BrdU^+Ki-67^−, red) or remaining active (BrdU^+Ki-67^+, purple) in the SGZ of *Atg7* cKO (TX) and control (Oil) P14 mice (*n* = 3 and 4 mice, Oil and TX, respectively). **e** Quantification of total RGLs (SOX2^+GFAP^+ RGLs/mm²) at P14 (*n* = 3 and 4 mice, Oil and TX, respectively). **f** Quantification of active RGLs entering quiescence (BrdU^+Ki-67^− RGLs/mm²) in the SGZ of *Atg7* conditional knockout (cKO; TX) and control (Oil) P21 mice (*n* = 8 and 7 mice, Oil and TX, respectively). **g** Percentage of P13 active RGLs (BrdU^+) entering quiescence (BrdU^+MCM2^−, red) or remaining active (BrdU^+MCM2^+, purple) in the SGZ of *Atg7* cKO (TX) and control (Oil) P21 mice (*n* = 5 and 5 mice, Oil and TX, respectively). **h** Quantification of quiescent RGLs that become active (BrdU^−Ki-67^+ RGLs/mm²) in the SGZ of *Atg7* cKO (TX) and control (Oil) P21 mice (*n* = 6 and 5 mice, Oil and TX, respectively). **i** Quantification of total RGLs (SOX2^+GFAP^+ RGLs/mm²) at P21 (*n* = 6 and 5 mice, Oil and TX, respectively). Data in panels (**c**–**i**) are presented as mean ± SEM. Statistics: unpaired *t*-test, two-tailed. *$p < 0.05$; **$p < 0.01$; ***$p < 0.001$. Representations shown in a. c. f. and h. were created with BioRender.com. Source data are provided as a Source Data file. TX, Tamoxifen. P3, Postnatal day 3. P4, Postnatal day 4. P5, Postnatal day 5. P14, Postnatal day 14. P10, Postnatal day 10. P13, Postnatal day 13. P21, Postnatal day 21. A, Active. Q, Quiescent. RGL, Radial Glia-Like Cell. BrdU, Bromodeoxyuridine. TX, tamoxifen.

percentage of proliferating P13 RGLs that were still entering quiescence at this later postnatal development phase in tamoxifen-treated animals (BrdU^+Ki-67^−/BrdU^+MCM2^− RGLs, Fig. 5f, g) and a concomitant increase in the activation of the already quiescent BrdU^− RGLs (Fig. 5h). We found no significant changes in the total number of RGLs (Fig. 5i), suggesting that proliferating P21 RGLs, as a population, are not dividing symmetrically and are not being lost through differentiation in the timeframe analyzed. Data were reproduced in independent in vivo treatments of different mouse litters (Supplementary Fig. 10c–e). As an additional control, we also evaluated *NesCreER^T2* mice (Supplementary Fig. 10c). We found no differences neither in the entry into quiescence

of the actively dividing BrdU^+ RGLs nor in the activation of the already quiescent BrdU^− RGLs when comparing oil *vs.* tamoxifen-treated animals (Supplementary Fig. 10f–h). Thus, the results indicate that *Atg7* is cell-intrinsically required for the proper acquisition and maintenance of RGL quiescence in postnatal weeks 1–3 when they are being spared as the dormant NSC reservoir that will persist throughout adulthood.

## Discussion

Despite the extensive literature on the intrinsic and extrinsic cues regulating NSCs in the adult mammalian brain[5], the mechanisms that cause the initial establishment of NSC quiescence during development

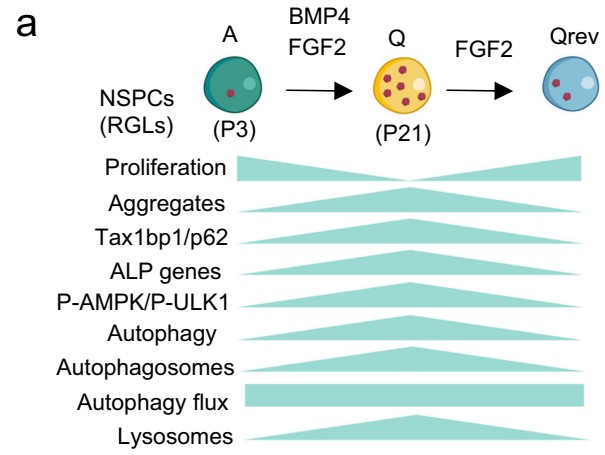

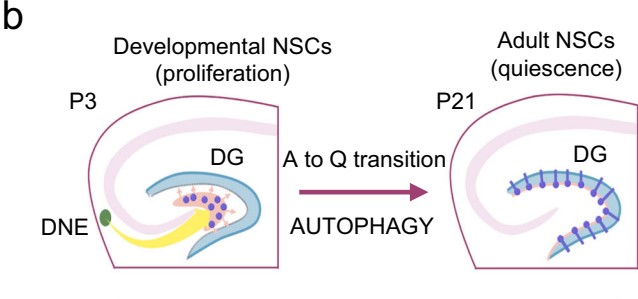

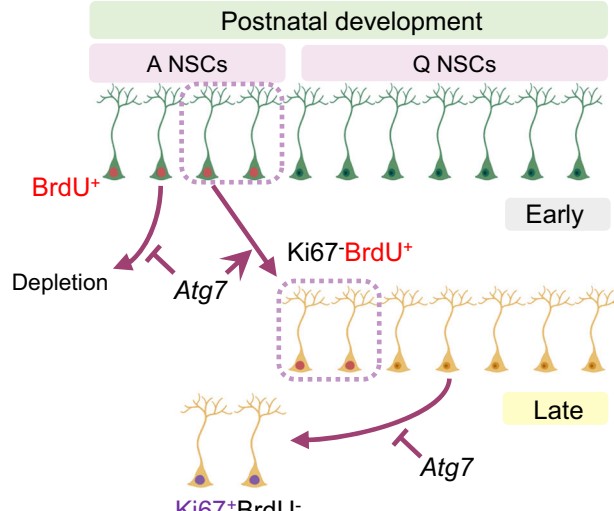

**Fig. 6 | Role of proteostasis and autophagy in the acquisition of NSC dormancy during development. a** Schematic representation illustrating the proposed protein aggregate dynamics and regulation of the autophagy-lysosomal pathway in the switch between the active (A) and the quiescent (Q) NSC states, both in vitro in NSPC cultures and in vivo in acutely isolated NSCs/RGLs. **b** Schematic representation illustrating the in vivo role of *Atg7*-dependent autophagy in quiescence acquisition and in the transition of developmental NSCs into their adult form. *Upper panel*, developmental dentate gyrus (DG) NSCs arise perinatally from NSCs that migrate out of the embryonic dentate neuroepithelium (DNE). Throughout postnatal development, they acquire a radial glia-like (RGL) morphology and they continue to proliferate until they transition to an adult-like quiescent state. *Lower panel*, *Atg7* is required early in dentate gyrus postnatal development for the acquisition of NSC/RGL quiescence. It is also required later on for the maintenance of RGL quiescence during the postnatal period, precluding the reactivation of the cells. The genetic strategy employed demonstrates that the autophagy gene *Atg7* is cell-intrinsically required for the proper regulation of quiescence in RGLs when they are being spared as a dormant NSC reservoir. Representations shown in (**a** and **b**) were created with BioRender.com. A, Active NSPCs. Q, Quiescent NSPCs. Qrev, quiescence-reverted NSPCs. P3, Postnatal day 3. P21, Postnatal day 21. ALP, Autophagy–lysosomal Pathway. NSCs, Neural Stem Cells. DG, Dentate Gyrus. DNE, Dentate Neuroepithelium. BrdU, Bromodeoxyuridine.

remain largely unexplored. Pro-quiescence niche-derived extrinsic signals such as BMPs maintain adult NSCs in a resting state; however, developmental NSCs are set aside as dormant reservoirs before the assembly of a functional niche structure. In addition, most proliferating embryonic and postnatal NSCs do not persist in the mature brain, but those that are being set aside as quiescent, likely need to be equipped with efficient proteostatic mechanisms for proper quality control and turnover of cytoplasmic contents, to ensure their lifelong survival as non-mitotic cells. In this manuscript, we demonstrate that autophagy plays a fundamental cell-intrinsic mechanistic role in the developmental transition from NSC proliferation to quiescence.

We focused on the formation of the hippocampal dentate gyrus SGZ NSC niche, which is protracted compared to the development of other brain structures, occurring mainly after birth between postnatal day P3 and P21. We found that dentate gyrus SGZ radial glia-like NSCs experience a proteostatic shift during the postnatal switch from proliferation to quiescence. This shift is evidenced by a reduction in

protein synthesis, accumulation of protein aggregates, increased expression of ALP-related genes and higher lysosomal content, and it is recapitulated in vitro in NSPC cultures challenged with the quiescence-promoting signal BMP4 (Fig. 6a). We further demonstrate that selective ablation of *Atg7*, a key gene for autophagosome formation, in hippocampal radial glia-like NSCs at early and late postnatal stages compromises, respectively, the initial acquisition and the maintenance of quiescence during dentate gyrus morphogenesis (Fig. 6b). Thus, autophagy is critically required by RGLs at the time they are being spared as a dormant pool in the emerging SGZ niche.

The accumulation of protein aggregates and lysosomal structures experienced during development by hippocampal GLAST+PROM-1+ NSCs and RGLs, uncovered in this study, is shared with that reported for adult NSCs from the mammalian V-SVZ neurogenic niche[11] and for slowly dividing progenitors of the embryonic ganglionic eminence (GE)[24]. However, previous studies in adult NSCs and GE progenitors focused on lysosomal function, paying little attention to the autophagy phase of ALP[11,24]. FOXO3 regulation of a network of ALP genes was described in cultured NSCs[12]. A role for aggresomes in the clearance of protein aggregates during NSC activation[13] and increased proliferation in conditional knockouts of the lysosomal master regulator *Tfeb*[14] was also reported. In this regard, we have encountered discrepancies in the protein homeostasis network employed by the cells that could be attributed to niche or stage differences.

First, adult active V-SVZ NSCs exhibit increased expression of UPS-associated genes and higher proteasome activity;[11] by contrast, in the hippocampal NSPC cell culture model employed in this study we did not detect differences in proteasome activity between the active and quiescent states. We nevertheless found a clear and significant increase in a variety of autophagy markers in quiescent cells. The rise in ProteoStat signal and the accumulation of LC3-II, SQSTM/p62, P-AMPK and P-ULK1 along with the aggrephagy receptor Tax1bp1 and LAMP2 suggests that autophagy increases in Q-NSPCs possibly as a compensatory response to the protein aggregates. Indeed, AMPK/ULK1 activation by phosphorylation clearly indicates that Q-NSPCs are engaged in autophagy induction. At a global scale, in our cell culture conditions, A-NSPCs and Q-NSPCs possibly deal similarly with short-lived soluble proteins that are preferentially degraded through the UPS, while Q-NSPCs likely employ the autophagy pathway to cope with the degradation of long-lived protein aggregates that cannot be diluted through cell division. Uncovering the nature of the protein aggregates or exploring additional roles of the autophagy vesicles in other forms of selective autophagy, such as mitophagy or ERphagy, are beyond the scope of this study, which aimed at elucidating the functional impact

of autophagy in the developmental transition from NSC proliferation to quiescence.

Second, previous work showed that adult quiescent V-SVZ NSCs exhibit increased expression of lysosome-associated genes and enlarged lysosomes but less LC3 accumulation in response to inhibitors of lysosomal acidification, pointing to a slower degradation of autophagosomes by lysosomes[11]. Instead, in our system Q-NSPCs had higher content not only in lysosomes, as reported for adult V-SVZ NSCs, but also in many active components of the autophagy machinery, accumulated higher LC3-II levels in response to BafA1 and had a similar autophagic flux compared to A-NSPCs in the timeframe analyzed. A similar flow along the vesicular pathway but higher basal content in autophagic machinery may yield higher net autophagy levels per cell in the quiescent state. Since autophagic flux is an accepted proxy for autophagic degradation activity, our results are in accordance with a recent report that employed NSC cultures from the adult V-SVZ and SGZ[14], showing LC3 accumulation and higher cathepsin activity (which reflects enhanced lysosomal function) in quiescent cells.

Additional important issues arise when comparing our in vitro functional assays with previously published studies performed on adult NSC cultures that reached partially contradictory conclusions. Leeman et al. manipulated ALP with BafA1, which blocks lysosomal acidification and found decreased activation of the quiescent V-SVZ NSCs in the presence of EGF and FGF2[11]. On the contrary, Kobayashi et al. found increased activation of NSCs with BafA1, which was related to the blockade of activated P-EGFR degradation through the lysosome in an autophagy-independent manner[14]. The decreased proliferative activity upon lysosome accumulation in adult NSCs was linked mechanistically to EGF receptor endocytosis and endolysosomal degradation, not to the role of lysosomes in ALP[14]. In line with Kobayashi et al. [14], our cell culture results also support higher activation of Q-NSPCs with BafA1 treatment. Nevertheless, all the assays in the current study were performed in the absence of EGF (only FGF2 was used as a mitogen) and thus are likely unrelated to EGFR signaling. Importantly, compared to previous studies, our data uncover the instructive capacity of enhancing autophagy through AMPK/ULK1 signaling for the entrance into quiescence. We demonstrate that increasing AMPK activity with Phenformin is sufficient to promote cell cycle exit of proliferating NSPCs in an ULK1-dependent, SQSTM/p62-dependent, and therefore autophagy-dependent, manner. This instructive role does not require active BMP signaling through SMAD1/5/8 phosphorylation, one of the main pathways that maintains quiescence in the adult NSC niche[9,18].

The RNAseq data presented in this study for cultured NSPCs and hippocampal NSCs during development also demonstrate that stem cells engage in a transcriptional programme that enhances the expression of both autophagy and lysosomal genes when they transition from the proliferating developmental state to the quiescent adult state. Finally, the in vivo analysis of *Atg7* conditional knockout mice has allowed us to accurately examine the cell-intrinsic role of autophagy in NSCs of the developing hippocampal dentate gyrus at different stages. Developmental NSCs from the embryonic dentate neuroepithelium (DNE) reach the dentate primordium where they continue to proliferate perinatally until they transition to an adult-like quiescent state (Fig. 6b). Blocking *Atg7*-dependent autophagy in early perinatal NSCs, at a timepoint when NSC proliferation peaks, impairs the acquisition of the quiescent state and consequently the RGL/NSC pool is expanded. Deleting *Atg7* later on, at a time point when the morphological pattern and organization of the SGZ niche is emerging, interferes both with the entrance into quiescence and with keeping the already quiescent RGLs/NSCs on hold. Altogether our results unfold an important role for autophagy in the initial establishment and maintenance of radial glia-like NSC quiescence during dentate gyrus SGZ niche formation. The current data support a model wherein autophagy acts as a cell-intrinsic driver in the onset of RGL quiescence during early postnatal development and as a proteostatic pathway involved in the maintenance of the quiescent state once fully acquired at later postnatal stages (Fig. 6a, b). Both steps are key for hippocampal RGLs during SGZ development when they are being spared as the dormant adult stem cell reservoir, a prerequisite for the adequate preservation of neurogenesis throughout adulthood. In summary, our study highlights the involvement of proteostasis and autophagy in the acquisition of NSC dormancy during development. We largely extend recent findings in adult mice on the role of lysosomes in adult quiescent NSC reservoirs[11,14] and pave the way for further studies aiming to elucidate the mechanistic details of the developmental acquisition of quiescence by the privileged stem cells that populate the mature mammalian brain.

## Methods

### Mice

All experimental procedures were approved by the Institutional Animal Care and Use Committee of Instituto de Biomedicina de Valencia and by the Bioethical Committee of CSIC (protocol 2018/VSC/PEA/0053 and protocol 1458/2023). Mice were maintained in the IBV-CSIC facility on a 12h/12h light/ dark cycle under constant temperature (23 °C) with food and water provided ad libitum. Wild-type C57BL/6JRccHsd mice were purchased from Inotiv (https://www.inotivco.com/model/c57bl-6jrcchsd). Transgenic C57BL/6-Tg(Nes-cre/ERT2)4Imayo (*NesCreER^T2*) were described in Imayoshi et al. [22]. B6.Cg-ATG7tm1Tchi (*Atg7^Fl/Fl*) were described in Komatsu et al. [23]. RCE mice were obtained from Jackson Laboratory (Bar Harbor, ME) (STOCK Gt(ROSA)26Sortm1.1(CAG-EGFP) Fsh/Mmjax (#32037)). *NesCreER^T2* mice were crossed with *Atg7^Fl/Fl* or *RCE* mice to obtain *NesCreER^T2;Atg7^Fl/Fl* and *NesCreER^T2;RCE*.

For experiments with wild-type animals, postnatal (P3 to P21) C57BL/6JRccHsd male and female mice were used. For experiments with transgenic animals, postnatal (P3 to P21) NesCreER^T2;Atg7^Fl/Fl male and female mice were used. For control experiments, 21-day-old NesCreER^T2;RCE male and female mice, and 2-month-old NesCreER^T2;RCE and NesCreER^T2;Atg7^Fl/Fl male mice were used. Animals from the same litter were randomly assigned to the tamoxifen or Oil group. 0.2 mg tamoxifen was intraperitoneally injected into postnatal mice by one injection per day for two consecutive days (P3-P4) to induce *Atg7* deletion in *NesCreER^T2;Atg7^Fl/Fl* mice. 0.25 mg tamoxifen was intraperitoneally injected into postnatal mice by one injection per day for four consecutive days (P10-P13) to induce *Atg7* deletion (*NesCreER^T2;Atg7^Fl/Fl* mice) or GFP expression (*NesCreER^T2;RCE* mice). Then, 100 or 50 mg/Kg BrdU was intraperitoneally injected into postnatal mice by one injection at P5 or P13, respectively. For control experiments (Supplementary Fig. 9a–f), 2 mg tamoxifen was intraperitoneally injected into 2-month-old mice by one injection per day for five consecutive days to induce *Atg7* deletion or GFP expression. Tamoxifen-injected mice were housed in individual cages and sacrificed at the times indicated in each experiment. Validation of recombination after tamoxifen treatment was performed by PCR and qPCR (Supplementary Fig. 9b and h, respectively). DNA was extracted from the dentate gyrus of 2-month-old *NesCreER^T2 Atg7^Fl/Fl* mice after tamoxifen/oil treatment.

### MACS and FACS

Decapitation (P3 mice) and cervical dislocation (P21 mice) were used as euthanasia methods in MACS and FACS procedures. Hippocampi from postnatal day 3 or day 21 C57BL/6JRccHsd mice were dissociated using *Neural Tissue Dissociation Kit* (Miltenyi, 130-092-628) with gentle MACS Octo Dissociator (Miltenyi, 130-095-937). GLAST-positive cells were isolated by MACS. Cells were labeled with *Anti-GLAST* (ACSA1) *microBead Kit* (Miltenyi, 130-095-862) and magnetic separation was performed through MS column (Miltenyi, 130-042-102). Cells were attached to 100 µg/mL matrigel-coated dishes for further analysis. For hippocampal NSCs isolation, cells were labeled with GLAST (ACSA-1)-PE (1:50, Miltenyi, 130-118-483) and Anti-Prominin-1-APC (1:10, Miltenyi,

130-102-197) antibodies and were isolated by FACS (BD FACSAria™ III Cell Sorter). Compensations were done on single-color controls and gates were set on unstained samples. Dead cells were excluded by staining with DAPI (1 mg/mL). Sorted NSCs were collected directly into N2 medium and plated in 24-well matrigel coated dishes. After 1 h, NSCs were fixed with 2% paraformaldehyde and used for subsequent analysis. For hippocampal NSCs isolation of P14 *NesCreER^{T2};Atg7^{Fl/Fl}* mice treated with oil/tamoxifen, cells were labeled with GLAST (ACSA-1)-PE (1:50, Miltenyi, 130-118-483) and Anti-Prominin-1-APC (1:10, Miltenyi, 130-102-197) antibodies and were isolated by FACS (BD FACSAria™ III Cell Sorter). Sorted NSCs were collected directly into PBS and centrifuged. Pellet was stored at −80 °C until RNA extraction.

## Cell culture

For proliferation assays, we used rat Adult Hippocampal Neural Stem and Progenitor Cells (NSPC)[9]. NSPCs were maintained in N2 medium, DMEM/F-12(1:1) (Gibco) adding N2 Supplement (100×) (Gibco), with 10 ng/mL of human fibroblast growth factor 2 (FGF-2) (PeproTech), growing in neurospheres or 20 ng/mL of FGF2 growing in poly-ornithine (10 μg/mL)/laminin (5 μg/mL) (Sigma-Aldrich/Millipore) coated dishes[9]. For the quiescence induction, NSPCs were incubated with 50 ng/mL BMP4 (PeproTech) and 10 ng/mL of FGF2 during 4 days[9]. To revert the quiescence state, NSPCs were incubated again only with N2 medium and FGF2. To activate the autophagy-lysosome pathway NSCPs were treated with 2.5 mM of Phenformin (Sigma, 349887) for 2 h. To specifically inhibit ULK1 before Phenformin treatment, we treated NSPCs with 100 nM of ULK1 kinase inhibitor (Sigma-Aldrich, SBI-0206965) for 1 h. To inhibit the autophagy-lysosome pathway, NSPCs were treated with 50 or 100 nM of Bafilomycin (Sigma-Aldrich, B179) during 2 h. To inhibit BMP type I receptor, NSPCs were treated with 10 μM DMH1 (Biotechne, 4126) for 1 h. RPF-GFP-LC3 plasmid[20] was transfected into proliferating and quiescent NSPCs by electroporation (10 μg plasmid/10^6 cells, NEPA21 Super Electroporator). After electroporation, cells were treated with 100 nM Bafilomycin for 6 h.

## p62 (*Sqstm1*) siRNA

In this, 25,000 NSPCs were seeded in 24-well matrigel-coated dishes 24 h before transfection. HiPerFect® Tranfection Reagent (Qiagen, 301702) was used to transfect ON-TARGET plus SMART pool Rat Sqstm1 siRNA (Dharmacon, L-089365-02-0005). Briefly, 37.5 ng siRNA were diluted in 100 μL N2 medium and 3 μL of HiPerfect Tranfection Reagent were added and mixed by vortexing. To allow the formation of transfection complexes, the mix was incubated 10 min at room temperature. Finally, the complexes were added drop-wise onto the cells and the plate was gently swirled to ensure uniform distribution of the transfection complexes. Cells were grown in the presence of 10 ng/mL of FGF-2 and quiescence was induced with 50 ng/mL BMP4. After 72 h, cells were fixed and Ki67 immunocytochemistry was done. A control well treated with HiPerfect Tranfection Reagent only was used as reference.

## Immunostaining

Immunohistochemistry and immunocytochemistry were performed using standard procedures. Animals were anesthetized with pentobarbital (80 mg/Kg) and perfused with saline and 4% PFA followed by 2 or 16 h postfixation (postnatal and adult mice, respectively) in 4% PFA at 4 °C. Postnatal brains were embedded in 4% agarose solution prior to vibratome processing. The brains were coronally sectioned in a vibratome (Leica Microsystems VT-1200-S). The resulting 40 μm free-floating sections were collected sequentially generating antero-posterior reconstructions of the hippocampus conformed by 1 section every 320 μm of hippocampal structure. Stereology was performed by the analysis of at least 3 coronal sections, 40 μm each, separated 320 μm one to each other. Sampling started at the first appearance of the infrapyramidal blade of the dentate gyrus. Cultured

neurospheres were disaggregated and seeded in matrigel dishes before fixation with 2% paraformaldehyde (Panreac). Samples (tissue and cells) were incubated with blocking solution (10% Fetal Bovine Serum, 0.2% or 0.5% Triton-X100 (for nuclear or cytoplasmic epitope, respectively) in 0.1 M phosphate buffer). Primary antibodies used for the stainings are as follows: Ki67 (1:150, Abcam, ab15580), LC3B (1:150, ABclonal, A19665), Phospho-SMAD1 (Ser463/465)/ SMAD5 (Ser463/465)/ SMAD9 (Ser465/467) (D5B10) (1:500, Cell Signaling, #13820), Lamp2 (1:50, Biolegend, 108511), p62 (1:150, Abcam, ab56416), SOX2 (1:250, Gene Tex, GTX101507 and R&D, AF2018), GLAST biotin (1:50, Miltenyi, 130-119-161), Tax1bp1 (1:250, Novus, NBP3-15794), GFP (1:200, Aves-Lab, GFP-1010), GFAP (1:300, Sigma, G3893), BrdU (1:300, Abcam, ab6326), MCM2 (1:150, BD Biosciences, 610700). Treatment with 2 N HCl was required before BrdU detection. Secondary antibodies used were: Alexa Fluor 555 goat anti-mouse (1:500, Invitrogen, A31570), Alexa Fluor 633 goat anti-mouse (1:100, Invitrogen, A21052), Alexa Fluor 488 goat anti-mouse (1:500, Invitrogen, A21202), Alexa Fluor 488 donkey anti-rabbit (1:500, Invitrogen, A21206), Alexa Fluor 647 donkey anti-rabbit (1:500, Invitrogen, A31573), Cy3 donkey anti-rabbit (1:500, Jackson, 711-165-152), Alexa Fluor 555 goat anti-rat (1:500, Invitrogen, A21434), Alexa Fluor 488 donkey anti-chicken (1:500, Jackson, 703-546-155) and Cy2 Streptadivin (1:200, Invitrogen, 016-220-084). ProteoStat dye was employed following ProteoStat Aggresome Detection kit instructions (Enzo, ENZ-51035). OPP dye was performed following Click-iT® Plus OPP Protein Synthesis Assay Kits (Molecular Probes, C10456). After staining, cells and all sections were mounted and preserved with 50% Mowiol (Polysciences, 17951), 2.5% DABCO (Sigma, D2522). Images were acquired with a Leica Spectral SP8 confocal microscope with a 40x Oil objective. Images were analyzed with Fiji Image J Software.

## Western blot

Cell extracts were fractionated by SDS-PAGE and transferred to a polyvinylidene difluoride (PVDF) membrane following the manufacturer's protocol (Bio-Rad). Membranes were incubated with 5% BSA (Sigma) in TBST for 60 min. Primary and secondary antibodies used for the Western blot assays are as follows: LC3 (1:1000, Santa Cruz, sc-376404), p62 (1:500, Abcam, ab56416), P-AMPK (1:1000, Cell Signaling, 2535), total AMPK (1:1000, Cell Signaling, 2793), P-ULK (1:1000, Cell Signaling, 5869), total ULK (1:1000, Cell Signaling, 8054), P-Raptor (1:1000, Cell Signaling, 2083), total-Raptor (1:1000, Cell Signaling, 2280), Phospho-SMAD1 (Ser463/465)/ SMAD5 (Ser463/465)/SMAD9 (Ser465/467) (D5B10) (1:500, Cell Signaling, #13820), BMPR2 (1:500, Abcam, ab130206), BMPR1A (1:500, Abcam, ab264043), Actin (1:5000, Sigma-Aldrich, A5441), IRDye 680LT anti-mouse (1:5000, Licor, 925-68020), IRDye 800CW anti-rabbit (1:5000, Licor, 925-32211). Proteins were detected with LI-COR Odyssey and analyzed with Image Studio™ Lite. Uncropped and unprocessed scans of the most important blots are supplied in the Source Data file.

## Gene expression analysis by PCR and gel electrophoresis

RNA was extracted from cells. cDNA was obtained by reverse transcription (RT) employing the PrimeScrip RT Reagent kit (Takara). Gene expression was determined by quantitative polymerase chain reaction (qPCR) in a QuantStudio 5 (Applied Biosystems) using SYBR PremixEX Taq (2×) (Takara) and the corresponding forward and reverse primer for each gene. *Sdha* was used as the internal reference for normalization. Data were analyzed according to the $2^{-\Delta\Delta Ct}$ method. The primers employed in qPCR and PCR are shown in Supplementary Data 2. PCR or qPCR products were resolved in a 2% agarose gel electrophoresis and were visualized under a UV transilluminator.

## RNAseq

For the transcriptomic analysis, total RNA was extracted from active and quiescent NSPCs. cDNA libraries were prepared using the TruSeq

Stranded mRNA Library Preparation Kit (Illumina, San Diego, CA). Libraries were sequenced on the Illumina® NSQ 550 platform to generate 1 × 75 bp single-end reads using the Illumina® NSQ 500/550 Hi Output KT v2 (75 CYS) (Illumina, San Diego, CA), according to the manufacturer's protocol. Libraries were sequenced on average to a depth of 440 M reads per library. Reads were mapped against the rat reference genome Rn6 with HISAT2. Gene expression quantification was performed with HTSeq. Differential gene expression analysis was done with DESeq2 with an adjusted p-value cutoff of 0.05.

## Measurement of proteasomal activity

*Proteasome 20S Assay kit* (Sigma-Aldrich) was employed to measure chymotrypsin-like activity associated with the proteasome. Active and quiescent NSPCs were grown on poly-ornithine/laminin dishes and 10 μM of MG132 was used as control. Proteasome activity was normalized to cell number. For cell number calculations, cells were fixed and stained with crystal violet. $A_{590}$ values were next used to determine cell number by interpolating from a standard curve.

## Statistics and study design

Cell culture experiments were independently performed at least 3 times and *n* refers to the number of times the experiment was repeated. Acute cell isolations from dissected hippocampi by MACS and FACS were performed independently at least 3 times, employing pooled tissue from 3 to 7 postnatal mice in each isolation, and *n* refers to the number of times the isolation procedure was repeated. For the in vivo analysis *n* refers to the number of mice. G*Power calculations were employed to reduce the number of animals needed. Sufficient sample size was maintained to obtain a Type I error as low as 0.05 and a power as high as 0.8. All animal experiments were performed with identically handled littermate controls. Treatments in P5 to P14 animals were performed once (one litter) while treatments in P13 to P21 animals were performed twice (two independent litters treated with reagents that were also prepared separately at different times separated in time). All statistical tests and sample sizes are included in the figure legends. Measurements were taken from distinct samples. Normality was measured with the Shapiro–Wilks test. Normal data are shown as mean ± SEM and non-normal data are represented as median ± interquartile range. In all cases, p-values are represented as follows: ****$p < 0.0001$, ***$p < 0.001$, **$p < 0.01$, *$p < 0.05$. For normal data, all quantifications were statistically analyzed using either one-sample *t*-test (normalized data 2 groups), Student's *t*-test-2 tails (2 groups) or 2-way ANOVA test. For non-normal data, the Wilcoxon Signed Rank Test or Mann–Whitney test was employed. Statistical analysis was performed using GraphPad Prism 8. Subjects were randomly assigned to treatment groups.

## Reporting summary

Further information on research design is available in the Nature Portfolio Reporting Summary linked to this article.

## Data availability

The RNAseq data generated in this study have been deposited in the NCBI BioProject and BioSample databases under accession code Bioproject PRJNA666074. The Sequence Read Archive (SRA) experiments with the following accession numbers were used: SRX9200622, SRX9200623, SRX9200624, SRX9200625, SRX9200634, SRX9200642. Source data are provided with this paper.

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

## Acknowledgements

We thank all members of the Mira laboratory, Dr. Marçal Vilar and Prof. Dr. Chichung Lie for their discussions and helpful suggestions; J. Durban, R. Viana, M. Quesada, M. Heredia and M. Cano for support with

RNAseq, confocal microscopy, histology and animal care, respectively. This work was supported by the Spanish MICINN (grant no. PID2019-111225RB-I00) and PROMETEO/2018/055 grant from Generalitat Valenciana for H.M.; FPI fellowship BES-2016-077156 to I.C.-B.; APOSTD2019/058 from Generalitat Valenciana to L.C.-C.; and INVEST/2022/242 from Generalitat Valenciana to H.M. and C.F.-I.

## Author contributions

H.M. conceived the study. I.C.-B. and H.M. designed the experiments. I.C.-B., L.C.-C. and C.F.-I. performed the assays, analyzed the data and prepared the figures. I.C.-B., J.G.-N. and C.F.-I. conducted the postnatal confocal microscopy imaging experiments and analyzed the imaging data. P.S. was involved in the design and interpretation of the autophagy flux and cell signaling experiments. H.M. wrote the main manuscript text. All authors reviewed the manuscript.

## Competing interests

The authors declare no competing interests.
