## [Peer Review File · Nature Communications]

Autophagy drives the conversion of developmental neural stem cells to the adult quiescent stateEditorial Note: This manuscript has been previously reviewed at another journal that is not operating a transparent peer review scheme. This document only contains reviewer comments and rebuttal letters for versions considered at *Nature Communications*.

Reviewers' comments:

Reviewer #1 (Remarks to the Author):

In the revised manuscript authors carefully replied to our previous concerns and improved their manuscript. The new experiments shown in Figure 1 and the newly added in vivo data using P3-P4 animals support the relevance of Atg7-dependent autophagy for the establishment of quiescence during early post-natal brain development. The new experiments add value to the manuscript.

Furthermore, the authors now added a much more extensive and careful discussion of their findings in relation to previous work. Similarities and differences compared to existing literature are more clearly discussed.

The current work focuses on the role of autophagy in the transition to quiescence, differing from previous publications where the emphasis was rather on lysosomal activity as a regulator of neural stem cell quiescence.

The paper is experimentally sound; however, the lack of conceptual novelty still remains a setback of the current manuscript. Although results here presented are not fully redundant with work presented in previous papers, it is still rather close to what has been reported before.

Reviewer #2 (Remarks to the Author):

The authors have added in vivo data with postnatal day 3-4 animals to strengthen the causative link between autophagy and quiescence. Specifically, they use BrdU label retention in conditionally deleted Atg7 mice (to reduce autophagy) and show that RGLs do

not acquire quiescence at two different timepoints: P5 to P14 and P13 to P21. Having these new experiments with P3 animals to distinguish early and late postnatal development is informative. However, the number of mice and independent replicates, and whether these experiments are sufficiently powered, in this new *in vivo* data in Figure 5 remains of concern.

In Figure 5c, the data is plotted as the number of active RGLs entering quiescence (BrdU+Ki67-) with the data presented as mean \pm SEM from $n=3$ mice per group. However, in the study design, the authors state that all experiments were independently repeated at least 3 times. Same with Figure 5e, oil control where $n=3$. Therefore, it is difficult for this reviewer to understand how these experiments were repeated at least 3 times when there is only $n=3$ mice per group. If the $n>3$ mice per group as stated, it would be expected that there would be at least $n>9$ mice per group if the experiments were independently replicated at least three times. Because these data are essential to the crux of the study's conclusions, it is essential to $n=3$ refers to the number of replicates or the number of mice.

Reviewer #3 (Remarks to the Author):

Calatayud-Baselga and colleagues propose an instructive role for AMPK-induced autophagy and increased number of lysosomes in the quiescence entry of neural stem cells, focusing on the dentate gyrus. As quiescent state entry is accompanied by an accumulation of protein aggregates, the enhanced autolysosomal system is proposed to combat protein aggregates. Using an *in vitro* assay, where entry to quiescence is induced by BMP4 treatment, the authors show that activation of AMPK prevents cell cycle re-entry of NSCs, and blocks proliferation in the presence of the mitogen Fgf2, in an Ulk1-dependent manner, suggesting that autophagy is required for quiescence. This was also supported by experiments *in vivo*, reporting decreased number of cells entering quiescence at P14 in animals with conditional ablation of *atg7* in NSCs.

While the findings are convincing and interesting, they lack mechanistic insight on the interplay of autophagy with the cell cycle in these very specialized cells.

Comments:

1. Given the emerging literature on the lipidation of LC3 and other Atg8 family members on non-autophagic membranes, in pathways such as LC3-associated endocytosis (LANDO), it is

necessary to exclude that the enhanced LC3 signal represents autophagic vesicles. In Figure 2, using western blot and immunocytochemistry in the in vitro system, the authors should determine the degree of lipidation that remains after Ulk1 inhibition, both in the presence and absence of BafA1. This is important, as their in vivo confirmation relies on ablation of Atg7, a protein that is also required for non-canonical Atg8 lipidation.

2. Does Ulk1 inhibition by itself prevent quiescence and promotes proliferation of NSCs in the absence of any exogenous signals (BMP4 and FGF2)?

3. It is unclear whether the aggregates themselves have an instructive role in the entry to quiescence. Several experiments can shed light into this question. First, the in vitro experiments monitoring transitions from active to quiescent states should be performed after knocking down p62 and other aggrephagy receptors that accumulate in the quiescent state. Second, protein aggregates should be isolated as insoluble fractions and analyzed by mass-spec to determine their composition.

4. Are protein aggregates the only cellular constituents that accumulate in the quiescent state? Given several selective autophagy pathways for different organelles, the authors should also monitor the accumulation of mitochondria, ribosomes and ER.

5. Does autophagy induction render NSCs unresponsive to BMP4 signaling? For example, the surface levels of BMP receptors 1 and 2 should be monitored upon treatment with the Ulk1 inhibitor.

6. In the experiments shown in Figure 1, approx. 25% of NSCs at P3 are proliferating and Ki67+. Do the differences in Lamp2-content reflect the proliferative state of these cells? In other words, are the lamp2-low cells of P3 and P21 both quiescent? Or are we simply comparing active P3 to quiescent P21 cells? How would they compare if quiescent and active cells of P3 were compared?

Answer to Reviewer#1

Thank you very much for recognizing the interest and quality of our work. Please let me take the opportunity to carefully explain the relevance of our findings and to highlight the conceptual advance provided, compared to the results presented in previously published papers.

Our data are of conceptual advance as we prove in a solid way the following novel findings:

1. The autophagy gene *Atg7* is cell-intrinsically required for the initial acquisition of RGL quiescence, as demonstrated through in vivo analysis of *Atg7* conditional knockout mice. This is a relevant finding since it is currently unknown how RGLs are first set aside during development as a dormant stem cell pool. We accurately demonstrate this main finding employing a cell type- and time-specific strategy, focusing on early developmental stages (not previously reported).
2. The autophagy gene *Atg7* is required in addition for maintaining RGLs in the quiescent state, once it has been fully acquired, at later postnatal stages (not previously reported).
3. In line with the above, protein synthesis decreases while protein aggregates accumulate in RGLs during development at the time they are being spared as a quiescent reservoir (not previously reported).
4. RGLs concomitantly engage in a transcriptional programme (in vivo and in vitro RNAseq data) that increases a variety of autophagy and autophagy machinery components (in vivo and in vitro cell biology data at the mRNA/protein level) (not previously reported in RGLs).
5. AMPK/ULK1 activation by phosphorylation clearly indicates that Q-NSPCs undergo autophagy induction (not previously reported), while proteasome activity is similar in Q-NSPCs and A-NSPCs (thus, hippocampal Q-NSPCs do not experience a delayed autophagy-lysosome flux and have less proteasome activity; opposite results to Leeman et al¹⁰ in adult V-SVZ).
6. Enhancing autophagy in NSPCs (through AMPK/ULK1 signaling) is instructive for the entrance into quiescence (not previously reported and conceptually opposite to the findings in Leeman et al¹⁰, who claimed that enhancing autophagy promotes entrance into the cell cycle).
7. Blocking the autophagy-lysosomal pathway activates quiescent NSPCs (contrary to Leeman et al¹⁰) even in the absence of EGF ligand and in an autophagy-dependent manner (not previously reported and mechanistically different to Kobayashi et al¹³, who linked the effect of blocking the autophagy-lysosomal pathway in adult NSCs to P-EGFR degradation through endolysosomes in an autophagy-independent manner).

Furthermore, in this reviewed version of the manuscript we also include the following new findings:

8. Increased lipidation of LC3 in quiescent NSPCs represents autophagic vesicles and not other non-autophagic membranes in pathways such as LC3-associated endocytosis (LANDO) (see new

Extended Data Fig. 3c-d western blot showing the degree of lipidation that remains after Ulk1 inhibition, both in the presence and absence of BafA1).

9. Blocking autophagy through Ulk1 inhibition in the presence of mitogenic signaling does not further promote proliferation of NSPCs (see new data in Fig. 4c).

10. Silencing SQSTM1/p62 (an autophagy receptor that accumulates in the quiescent state) by siRNA impairs the entry to quiescence (Fig. 4b), shedding light to the instructive role of aggregates themselves in the entry to quiescence.

11. At a mechanistic level, we have monitored nuclear P-Smad1/5/9 (as a readout of canonical BMP signaling) and we have performed new functional experiments employing the selective BMP receptor inhibitor DMH1. We previously determined that P-Smad signaling downstream of the BMPR1A/BMPR2 is involved in maintaining NSC quiescence in the adult hippocampal stem cell niche (Mira et al., Cell Stem Cell 7, 78-89 (2010)). Our results now show that the instructive role of autophagy in quiescence acquisition is not mediated by canonical BMP signaling, one of the main pathways that keeps NSCs out of the cell cycle during adulthood. The results are presented in Extended Data Fig. 7 and Fig. 8. We have also determined BMP type 1 and Type 2 receptor levels (BMPR1A/BMPR2) in NSPCs upon inducing and blocking autophagy following SBI treatment (Ulk1 inhibitor) and Phenformin treatment (AMPK activator). We have found no significant differences (see Extended Data Fig. 8).

Thus, the current work:

- a) focuses on the role of autophagy in the initial developmental transition from proliferating NSCs into and their adult state (please see our new graphical abstract in figure 5 and the new text in the discussion section, line 305 onwards),

Figure 5k. Schematic representation illustrating the *in vivo* role of *Atg7*-dependent autophagy in quiescence acquisition and transition of developmental NSCs into their adult form. *Upper panel*, developmental dentate gyrus (DG) NSCs arise perinatally from NSCs that migrate out of the embryonic dentate neuroepithelium (DNE). Throughout postnatal development, they acquire a radial glia-like (RGL) morphology and they continue to proliferate until they transition to an adult-like quiescent state. *Lower panel*, *Atg7* is required early in dentate gyrus postnatal development for the acquisition of NSC/RGL quiescence. It is also required later on for the maintenance of RGL quiescence during the postnatal period, precluding the reactivation of the cells. The genetic strategy employed demonstrates that the autophagy gene *Atg7* is cell-intrinsically required for the proper regulation of quiescence in RGLs when they are being spared as a dormant NSC reservoir.

- b) differs from previous publications where the emphasis was rather on lysosomal activity as a regulator of adult neural stem cell quiescence,

- c) demonstrates that autophagy *per se*, through AMPK/ULK1/SQSTM1/p62/Atg7, is a fundamental regulator of NSC/RGL dynamics in the developing hippocampal dentate gyrus,
- d) shows that the autophagy branch of ALP acts as an instructive mechanism for the entrance into quiescence, that is operating during niche development to allow for the adequate number of RGLs to be spared as dormant reservoir.

We think we adequately clarify the original contribution of our work in the article resubmitted to Nature Communications. The main shared finding with previous reports in adult NSC niches is simply the accumulation of protein aggregates and lysosomal structures in RGLs. Other than that, we have encountered marked discrepancies in how the protein homeostasis network is employed by developing RGLs as they acquire their adult identity. Once again, we would like to emphasize that our *in vivo* and *in vitro* data shed light for the first time on an intrinsic cellular (autophagic) mechanism that directs the initial establishment of the dormant neural stem cell identity in the mammalian brain during development. We have changed the manuscript title to better reflect this idea: ***“Autophagy drives the conversion of developmental neural stem cells to the adult quiescent state”***.

In summary, throughout the rounds of review we have received feedback from the referees, including very valuable recommendations and insights, and we have greatly reinforced some previous weaknesses. Especially we have focused on points that clearly distinguish our research at a conceptual level from published work. We have also solved other caveats related to the analysis, description, or interpretation of some of the data. All of the points raised by the reviewers have been addressed as they have been either incorporated or discussed and we have generated a greatly improved manuscript for Nature Communications with relevant, clear, and compelling results. Our study thus delves into the unsolved question of how developmental NSCs acquire an adult NSC state, which we show is a developmental process with defined proteostatic requirements. The active-to-quiescence / developmental-to-adult switch occurs before a functional adult niche structure is assembled. The cellular mechanisms underlying this fundamental process are interesting on their own, regardless of the knowledge previously achieved in adult niches. Please note this is currently a highly competitive topic in the stem cell field and world leaders are actively working to decipher the molecular and cellular events that promote this unexplored transition, since it is the establishment of the adult NSC pools what ensures the lifelong persistence of neuronal production in the mammalian (human) brain.

We most sincerely hope Reviewer#1 will appreciate our effort and will consider that the findings are convincing and provide sufficient insight on the cellular mechanisms that are required by developmental NSCs to finally come to rest and transition into their adult form.

Answer to Reviewer#2

We thank the reviewer for acknowledging that we have strengthened the causal mechanistic link between autophagy and the developmental transition of NSCs/RGLs into quiescence, by analysing BrdU label retention in conditionally deleted *Atg7* mice, employing a cell type-specific and stage-specific strategy at two different developmental timepoints.

Additional concerns raised by Reviewer#2 related to Figure 5 have now been addressed as follows:

1. *“The authors have added in vivo data with postnatal day 3-4 animals to strengthen the causative link between autophagy and quiescence. Specifically, they use BrdU label retention in conditionally deleted Atg7 mice (to reduce autophagy) and show that RGLs do not acquire quiescence at two different timepoints: P5 to P14 and P13 to P21. Having these new experiments with P3 animals to distinguish early and late postnatal development is informative. However, the number of mice and independent replicates, and whether these experiments re sufficient powered, in this new in vivo data in Figure 5 remains of concern.”*

The reviewer points out the “Reduced” number of animals analysed in Fig 5 for the P5 (BrdU) to P14 (sac.) study compared to the previously performed P13 (BrdU) to P21 (sac.) study. Please allow us to explain this discrepancy. The widely used G*Power free statistical software was employed to estimate the sample size for the new P5-P14 experiments. Calculation with G*Power was performed using experimental data from our first P13-P21 postnatal analysis (active (Ki67+) RGL data from Oil/tamoxifen treated groups provided in the 1st version of our manuscript submitted to Nature Neuroscience). Input parameters were set to obtain a Type I error as low as 0.05 and a power as high as 0.8 (80% probability the researcher will not commit a type II error, which by convention is an acceptable level of power). The G*Power result is shown in the following box:

Note that the estimated sample size for group 1 and group 2 is n=3. Following the 3 R's "Reduction" principle, this prompted us to use fewer animals in the new experimental design performed at P5-P14, thus limiting the unnecessary inclusion of additional animals. **The use of G*Power for the *in vivo* design is now explained in the study design section.**

2. "In Figure 5c, the data is plotted as the number of active RGLs entering quiescence (BrdU+Ki67-) with the data presented as mean \pm SEM from n=3 mice per group. However, in the study design, the authors state that all experiments were independently repeated at least 3 times. Same with Figure 5e, oil control where n=3. Therefore, it is difficult for this reviewer to understand how these experiments were repeated at least 3 times when there is only n=3 mice per group. If the n>3 mice per group as stated, it would be expected that there would be at least n>9 mice per group if the experiments were independently replicated at least three times. Because these data are essential to the crux of the study's conclusions, it is essential to n=3 refers to the number of replicates or the number of mice."

We thank the reviewer for the comment and we apologize for the lack of clarity in the previous version of the manuscript regarding the number of replicates and the number of mice. We also fully understand the reviewer's concern regarding the *in vivo* data, which we agree "are essential to the crux of the study's conclusions".

Given the logical concern about the reproducibility of the *in vivo* treatments, we had already performed an independent treatment in P13 to P21 mice prior to submission (designed as n=3 according to G*Power). However, the results were not included in the paper. The data showed that the *in vivo* results are fully reproducible in independent treatments, performed at a different time, with completely independent animal litters, using reagents that were also prepared separately. As mentioned above, this material was available before but we thought it was not necessary to provide these additional results. In light of the Reviewer's comment, we are now presenting them in **Extended data Figure 8 f-h**. Data from these 3 OIL and 3 TX treated mice are not included in Figure 5 plots.

We have also corrected the wording of the section related to the study design in the Online methods file as follows:

"Online Methods – Statistics and study design

Cell culture experiments were independently performed at least 3 times and *n* refers to the number of times the experiment was repeated. Acute cell isolations from dissected hippocampi by MACS and FACS were performed independently at least 3 times, employing pooled tissue from 3-7 postnatal mice in each isolation, and *n* refers to the number of times the isolation procedure was repeated. For the *in vivo* analysis *n* refers to the number of mice. G*Power calculations were employed to reduce the number of animals needed. Sufficient sample size was maintained to obtain a Type I error as low as 0.05 and a power as high as 0.8. All animal experiments were performed with identically handled littermate controls. Treatments in P5 to P14 animals were performed once (one litter) while treatments in P13 to P21 animals were performed twice (two independent litters treated with reagents that were also prepared separately in time)."

We have also corrected the wording of the section related to the **Life sciences study design** in the **Reporting Summary** file as follows:

Sample size: For the *in vitro* experiments, sample sizes (*n*) were not predetermined but are similar to those reported in previous publications. For the *in vivo* experiments G*Power calculations were employed to reduce the number of animals needed. Sufficient sample size was maintained to obtain a Type I error as low as 0.05 and a power as high as 0.8. For all *in vitro* determinations, the sample size was $n \geq 3$. For all *in vivo* determinations, the number of analysed mice was $n \geq 3$, except for Extended Data Figure 9. The exact sample size for each analysis is given as a discrete number in the Statistics file.

Data exclusions: Data were not excluded from relevant analysis unless GraphPad Prism identified outliers for normal distributions.

Replication: All *in vitro* experiments were independently repeated at least 3 times. The number of repetitions are listed in each figure. For the *in vivo* experiments, independent Oil and tamoxifen treatments of littermates were performed once at P3-P5 (for the *NesCreER^{T2};Atg7^{F/FI}* animals sacrificed at P14), twice at P10-P13 (for the animals *NesCreER^{T2};Atg7^{F/FI}* sacrificed at P21), once at P10-P13 (for the control animals *NesCreER^{T2};RCE* sacrificed at P21) and once at 2 months for the *NesCreER^{T2};RCE* and *NesCreER^{T2};Atg7^{F/FI}* animals included in Extended Data Figure 9.

Randomization: The order of samples to perform experiments was randomized for each experiment. Animals were randomly assigned to treatment groups.

Blinding: All experiments were performed as blindly as possible. However when blinding was not possible, data were collected and analysed without bias."

We hope the new data provided and the changes in the text satisfy the reviewer's concern.

In summary, our study thus delves into the unsolved question of how developmental NSCs acquire an adult NSC state, which we show is a developmental process with defined proteostatic requirements. The active-to-quiescence / developmental-to-adult switch occurs before a functional adult niche structure is assembled. The cellular mechanisms underlying this fundamental process are interesting on their own, regardless of the knowledge previously achieved in adult niches. Please note this is currently a highly competitive topic in the stem cell field and world leaders are actively working to decipher the molecular and cellular events that promote this unexplored transition, since it is the establishment of the adult NSC pools what ensures the lifelong persistence of neuronal production in the mammalian (human) brain.

Throughout the rounds of review we have received feedback from the referees, including very valuable recommendations and insight, and with their help we have greatly reinforced some previous weaknesses. Especially we have focused on points that clearly distinguish our research at a conceptual level from published work. We have also solved other caveats related to the analysis, description, or interpretation of some of the data. All of the points raised by the reviewers have been addressed as they have been either incorporated or discussed and we have generated a greatly improved manuscript for Nature Communications with relevant, clear, and compelling results.

We most sincerely hope the Reviewer#2 will appreciate our effort and will consider that the findings are convincing and provide sufficient insight on the cellular mechanisms that are required by developmental NSCs to finally come to rest and transition into their adult form.

Answer to Reviewer#3

We are very grateful to Reviewer#3 for the all the suggested additional experiments, which have greatly reinforced our work. Please let us take the opportunity to carefully answer to the new short list of concerns:

1. Given the emerging literature on the lipidation of LC3 and other Atg8 family members on non-autophagic membranes, in pathways such as LC3-associated endocytosis (LANDO), it is necessary to exclude that the enhanced LC3 signal represents autophagic vesicles. In Figure 2, using western blot and immunocytochemistry in the *in vitro* system, the authors should determine the degree of lipidation that remains after Ulk1 inhibition, both in the presence and absence of BafA1. This is important, as their *in vivo* confirmation relies on ablation of Atg7, a protein that is also required for non-canonical Atg8 lipidation.

We thank the reviewer for this suggestion. As requested, we have performed the additional control experiments. In **Extended data Figure 3** we now show the degree of LC3 lipidation that remains after ULK1 inhibition with SBI. We have used western blot analysis which we believe is the preferred technique for this type of determination, since it is easy to discriminate between the lipidated and non-lipidated LC3 forms. Quantification of the signal indicates that lipidated LC3II accumulation in Q-NSPCs is ULK1-dependent, validating that the enhanced signal in quiescent cells accounts for an increase in autophagic vesicles. The LC3 II / LC3 I ratio reflects the degree of LC3 lipidation and was also ULK1-dependent both in the presence and absence of BafA1.

We comment on these finding in the main text (highlighted in red):

“Enhanced LC3-II levels in Q-NSPCs accounted for an increase in autophagic vesicles, since quantification of the signal in the presence of the selective unc-51 like autophagy activating kinase 1 (ULK1) inhibitor (SBI-0206965) showed that lipidated LC3-II accumulation was dependent on the initiation of the autophagy pathway (Extended Data Fig. 3c-d)..... These findings suggest that the increase in ALP markers in quiescent cells is likely due to an increase in autophagy machinery content, rather than due to a deficient autophagic flux or an increase in non-autophagic membranes related to pathways such as LC3-associated endocytosis.”

We hope the new data provided and the changes in the text satisfy the reviewer's concern.

2. Does *Ulk1* inhibition by itself prevent quiescence and promotes proliferation of NSCs in the absence of any exogenous signals (BMP4 and FGF2)?

Unfortunately, it is not possible to perform proliferation/quiescence NSC assays in vitro in the absence of exogenous signals, since upon removing the mitogen NSCs readily differentiate (thus producing a completely different biological scenario). However, to get as close as possible to the Reviewer's concern, we have performed ULK1 inhibition experiments in FGF2-only (minimal medium to maintain stemness). As we now show in **Figure 4c**, ULK1 inhibition (blocking autophagy initiation) by itself does not further promote proliferation of NSCs in the absence of exogenous BMP4 (+SBI vs control condition (-Phen -SBI); grey bar vs green bar):

We comment the following in the text:

*“Our results demonstrated that instruction of the quiescent state upon increasing AMPK activity was ULK1-dependent (Fig. 4c). **ULK1 inhibition by itself did not raise proliferation of the A-NSPCs (Fig. 4c).** hope the new data provided and the changes in the text satisfy the reviewer's concern.”*

3. It is unclear whether the aggregates themselves have an instructive role in the entry to quiescence. Several experiments can shed light into this question. First, the in vitro experiments monitoring transitions from active to quiescent states should be performed after knocking down p62 and other aggregophagy receptors that accumulate in the quiescent state. Second, protein aggregates should be isolated as insoluble fractions and analyzed by mass-spec to determine their composition.

As requested by Reviewer#3, we have performed additional experiments. In **Figure 4b** we now show that, as expected, knocking down SQSTM/p62 by siRNA impairs the entry into quiescence:

Regarding the second comment, we still feel that shedding light into the proteomic composition of detergent-insoluble protein aggregates, although interesting, is really beyond the current developmental focus of the work.

We hope the new data provided and the changes in the text satisfy the reviewer's concern, and we also hope the reviewer will understand our point of view regarding the mass-spectrometry experiments to determine aggregate composition.

4. Are protein aggregates the only cellular constituents that accumulate in the quiescent state? Given several selective autophagy pathways for different organelles, the authors should also monitor the accumulation of mitochondria, ribosomes and ER.

We had already monitored the accumulation of mitochondria and ribosomes in active and quiescent cells for a different study. The data are included as figures in a collaborative manuscript that is in preparation. As you can see in the figure below, quiescent NSPCs have more mitochondria (panel A) but lower ribosomal content (panel B) compared to active NSPCs. Mitochondria accumulation is due to increased biogenesis through PGC1a activity (unpublished data, not shown) while the lower ribosomal content is likely linked to the decreased protein synthesis activity. As requested by the reviewer, we have now estimated endoplasmic reticulum (ER) content by immunofluorescence using antibodies against the integral ER protein Calnexin (CNX) (panel C). The data indicate that quiescent NSPCs may have a slightly higher ER content compared to active NSPCs, although the difference is not significant. Therefore, not all cellular constituents accumulate in the resting state; it is not a general rule. We agree selective autophagy pathways can potentially modulate the content of many cellular constituents in NSPCs, however we feel that these data are somewhat tangential to the core of our study and we have not included them in the article. We nevertheless show the results here to answer the reviewer's concern.

5. Does autophagy induction render NSCs unresponsive to BMP4 signaling? For example, the surface levels of BMP receptors 1 and 2 should be monitored upon treatment with the Utk1 inhibitor.

We thank the reviewer for suggesting these new experiments which we think are very interesting. We previously determined that the Bmpr1a/Bmpr2/P-Smad canonical signaling is involved in maintaining NSC quiescence in the adult hippocampal stem cell niche (Mira, H. et al. Signaling through BMPR-IA regulates quiescence and long-term activity of neural stem cells in the adult hippocampus. *Cell Stem Cell* 7, 78–89 (2010)).

At a mechanistic level, we have now monitored nuclear P-Smad1/5/9 (as a readout of canonical BMP signalling) and we have performed new functional experiments employing the highly selective BMP type I receptor inhibitor DMH1. Our results show that inducing or blocking autophagy following SBI treatment (Ulk1 inhibitor) and Phenformin treatment (AMPK activator) do not enhance or reduce nuclear P-Smad levels (Extended Data Fig. Fig. 8). BMP4 stimulation was used as a positive control.

As for the DMH1 treatment, the results indicate that the instructive role of autophagy in quiescence acquisition is not mediated by canonical BMP signaling (Extended Data Fig. 7).

We have also determined BMP type 1 and type 2 receptor levels (BMPR1A/BMPR2) in NSPCs upon inducing and blocking autophagy following SBI treatment and Phenformin treatments. We have found no significant differences BMPR1A/BMPR2 levels (see Extended Data Fig. 8), indicating that autophagy does not render NSPCs unresponsive to BMP4 signalling.

We comment on these findings in the main text, results section:

“Interestingly, promotion of the quiescent state through autophagy induction did not require canonical BMP signalling. As shown in Extended Data Fig. 7b-e, the highly selective BMPR inhibitor DMH1 had no significant effect on the acquisition of NSPC quiescence induced by Phen, although it completely blocked cell cycle exit following BMP4 stimulation. Phosphorylation of the canonical BMP effector proteins SMAD1/5/9 in NSPCs and their nuclear translocation was unaffected by autophagy induction and inhibition (Phen and SBI treatments, respectively; Extended Data Fig. 8). The levels of the type I and type II BMP receptors BMPR1A and BMPR2 were not significantly affected (Extended Data Fig. 8a-b). Thus, our results demonstrated that

quiescence instruction by AMPK activation with Phen was not mediated by canonical BMP signalling.”

6. In the experiments shown in Figure 1, approx. 25% of NSCs at P3 are proliferating and Ki67+. Do the differences in Lamp2-content reflect the proliferative state of these cells? In other words, are the lamp2-low cells of P3 and P21 both quiescent? Or are we simply comparing active P3 to quiescent P21 cells? How would they compare if quiescent and active cells of P3 were compared?

Lamp2 content in active and quiescent NSCs in vivo was already measured at two postnatal timepoints (P14 and P21) in which Ki67+ and Ki67- RGLs can be unambiguously distinguished based on their morphology. Indeed, Lamp2 levels in active (Ki67+) NSCs were significantly lower than in quiescent (Ki67-) NSCs (see results in **Figure 1h** by confocal analysis of tissue sections). This clearly indicates that the differences in Lamp2 content reflect the proliferative state of postnatal NSCs. We have included the conclusion in the results section of main text to state it explicitly: *“LAMP2 content reflected the proliferative state of the cells, since LAMP2-low RGLs were actively proliferating while LAMP2-high cells were quiescent.”* We have also modified the labels in Figure 1h to make this more clear.

In summary, throughout the rounds of review we have received feedback from the referees, including very valuable recommendations and insight, and with their help we have greatly reinforced some previous weaknesses. Especially we have focused on points that clearly distinguish our research at a conceptual level from published work. We have also solved other caveats related to the analysis, description, or interpretation of some of the data. All of the points raised by the reviewers have been addressed as they have been either incorporated or discussed and we have generated a greatly improved manuscript for Nature Communications with relevant, clear, and compelling results.

We most sincerely hope the Reviewer#3 will appreciate our effort and will consider that the findings are convincing and provide sufficient insight on the cellular mechanisms that are required by developmental NSCs to finally come to rest and transition into their adult form.

REVIEWERS' COMMENTS

Reviewer #2 (Remarks to the Author):

Thank you for the detailed explanation of the study design (e.g., use of G*power calculations) and clarifying the sample sizes, which overall strengthens the conclusions and confidence of the findings.

Reviewer #3 (Remarks to the Author):

The authors have sufficiently addressed most of my concerns and comments, having performed additional experiments or by explaining their limitations in doing so.

One issue that remains relates to the experiments they performed in response to my first comment: Whether ULK1 inhibition results in a significant reduction in LC3 lipidation. The findings presented in extended data Figure 3c lack statistical analysis and any indication of the number of replicates used. As such, they cannot be used to conclude that LC3 lipidation is ULK1-dependent. I urge them to finalize this analysis/figure in order to support this claim.

Response to Reviewer #3

We are grateful to Reviewer#3 for the final comment. We have finalized Figure 3c (n=3 independent experiments) and we now provide the requested statistical analysis. We show that ULK1 inhibition in quiescent cells results in a significant reduction in LC3 lipidation.